# Activation Map Compression through Tensor Decomposition for Deep Learning

**Le-Trung Nguyen**     **Aël Quélennec**     **Enzo Tartaglione**
**Samuel Tardieu**     **Van-Tam Nguyen**

LTCI, Télécom Paris, Institut Polytechnique de Paris
{name.surname}@telecom-paris.fr

## Abstract

Internet of Things and Deep Learning are synergetically and exponentially growing industrial fields with a massive call for their unification into a common framework called Edge AI. While on-device inference is a well-explored topic in recent research, backpropagation remains an open challenge due to its prohibitive computational and memory costs compared to the extreme resource constraints of embedded devices. Drawing on tensor decomposition research, we tackle the main bottleneck of backpropagation, namely the memory footprint of activation map storage. We investigate and compare the effects of activation compression using Singular Value Decomposition and its tensor variant, High-Order Singular Value Decomposition. The application of low-order decomposition results in considerable memory savings while preserving the features essential for learning, and also offers theoretical guarantees to convergence. Experimental results obtained on main-stream architectures and tasks demonstrate Pareto-superiority over other state-of-the-art solutions, in terms of the trade-off between generalization and memory footprint.[1]

## 1   Introduction

Recent advances in Deep Learning have enabled Deep Neural networks to be used as an efficient solution for a wide variety of use cases, including computer vision [21, 39, 30], speech recognition [8, 31] and natural language processing [45, 42]. Much of this performance improvement is linked to the exponential increase in the number of parameters in the neural architectures. According to Sevilla *et al.*, the release of AlphaGo [47] in late 2015 marks the advent of a new era, which they call the "Large Scale Era", in reference to the computational cost of training doubling every 8 to 17 months [38]. At the root of exponential growth in neural network size is the improvement in hardware capabilities, particularly those designed for large-scale parallel computing, such as GPUs and TPUs [1]. While this trend demonstrates the strength of neural networks as a powerful generalization tool in many fields, it goes in the opposite direction when it comes to environmental concerns [41], making the deployment of newer architectures increasingly difficult. This is particularly true for edge devices such as mobile phones and embedded sys-

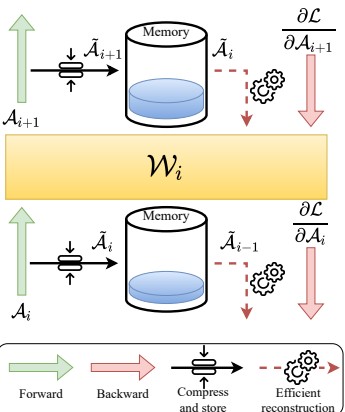

Figure 1: We compress the activations that will later be employed for backpropagation.

[1]Code: https://github.com/Le-TrungNguyen/NeurIPS2024-ActivationCompression.git

38th Conference on Neural Information Processing Systems (NeurIPS 2024).

tems, which cannot afford the high computing or memory costs. To address these challenges, three interconnected factors must be taken into account: power consumption, memory usage, and latency.

When considering larger neural networks with more layers and nodes, reducing their storage and computational cost becomes essential, especially for certain real-time applications such as edge computing. In addition, recent years have seen significant advances in the fields of intelligent edge and embedded intelligence, creating unprecedented opportunities for researchers to address the fundamental challenges of deploying deep learning systems on edge devices with limited resources (e.g. memory, CPU, power, bandwidth). Efficient deep learning methods can have a significant impact on distributed systems and embedded devices for artificial intelligence. Since training is supposed to take place in the cloud, most research on model compression and acceleration is specifically focused on inference [7]. There is, however, an emerging area of research concerning on-device training, which represents a decisive advance in the field of artificial intelligence, with considerable implications for a variety of practical situations [17]. Models trained offline on a dataset built at one point in time tend to fall victim to data drift when deployed "in the wild" [36]. Its combination with online learning strategies has the potential to enable continuous model improvement after deployment [14], thus adapting the model predictions to observed evolutions in the data distribution. We can illustrate this with the example of sensors in autonomous vehicles, where deep learning models must correctly classify vehicles with new designs never seen in the training set. Other advantages of on-device learning include security and privacy. By processing data locally, sensitive information remains more secure and less susceptible to data breach, a major concern in applications such as healthcare.

The main challenge limiting the feasibility of on-device learning lies in the computational demands of the backward pass, as gradient computation and parameter updates are significantly more resource-intensive than the forward pass (Appendix A.1). On embedded devices, memory and computation limitations act as strict budgets that must not be exceeded. Some approaches address the memory constraints by exploring alternatives to traditional backpropagation, including unsupervised learning for image segmentation [51], the Forward-Forward algorithm [16], and PEPITA [34]. While promising, these methods typically fall short of backpropagation-based techniques in terms of performance. A pioneering effort by Lin *et al.*, demonstrated that fine-tuning a deep neural network within a 256 kB of memory is feasible by selectively updating a sub-network, achieving a good results [25]. In a complementary approach, Yang *et al.* proposed reducing the number of unique elements in the gradient map through patch-based compression of the input and gradients of a given layer with respect to the output, thereby lowering memory costs and speeding up the learning process [53].

Inspired by tensor decomposition methods, we propose a method that compresses activation maps, reducing the memory demands for backpropagation while maintaining the deep neural network's generalization capability(Fig. 1). Our approach adaptively captures the majority of tensor variance, offering guaranteed accuracy in the gradient estimation. The key contributions of our work are as follows:

- We propose to exploit powerful low-rank approximation algorithms to compress activation maps, enabling efficient on-device learning with controlled information loss (Sec. 3.2 and Sec. 3.3).
- We provide a theoretical foundation for our method, along with an error analysis demonstrating that high compression ratios are achievable with limited performance degradation (Sec. 3.4).
- We extensively explore a diverse experimental landscape, demonstrating the generalization capacity of our proposed algorithm (Sec. 4).

## 2 Related Works

**Tensor Decomposition.** Model compression and acceleration is a well-developed and ongoing area of research in the deep learning community. It can be divided into two major areas, namely hardware and algorithm design. In the case of algorithmic compression, the five main components that stand out are efficient and compact model design, data quantization, network sparsification, knowledge distillation, and tensor decomposition [4, 7]. All these approaches aim to reduce the space occupied by network parameters. Also known as a low-rank approximation, *tensor decomposition* has emerged as a robust solution due to its combination of grounded theoretical findings and its

practicality in terms of hardware implementation [49]. Originating in the field of systems theory and signal processing [28], low-rank approximation has attracted growing interest in the deep learning community. Early examples were limited to the application of singular value decomposition (SVD) used to compress fully connected layers [50]. More recent work includes the application of generalized Kronecker product decomposition (GKPD) to both linear and convolutional layers [13], the introduction of semi-tensor product (STP) to improve compression ratios with reduced loss [56] or even more recently the acceleration of inference and backpropagation in vision transformers [52]. Evolutions in this domain can be summed up as improvements in compression ratios, latency, and power consumption, with limited performance loss and diversification of architectures considered.

**Activation Map Compression.** The term "activations" here refers to the outputs of each layer after the non-linearity has been applied, which are then fed to the next layer. While model compression generally aims at reducing the storage space occupied by parameters (which typically translates into network acceleration during inference), a key observation is that activations occupy much more space than parameters in memory during backward pass, as they are required to compute weight derivatives [2]. This state of facts motivates a new line of research, whose main objective is to compress activation maps using methods inspired by the literature on weight compression. For example, we mention the use of quantization [9], sparsification [22], a combination of both with the addition of entropy encoding [12], or even the application of wavelet transform in combination with quantization [10]. With the exception of Eliassen *et al.*'s work which accelerates training runtime, most of these works focus on accelerating inference, in a similar way to traditional model compression.

**On-device Learning.** Today's typical pipeline for on-device AI consists of designing resource-efficient deep neural networks [24], training them offline on target data, compressing them, and deploying them for inference on the resource-constrained environment. Alongside the rapid commercial expansion of the IoT field and the ubiquitous presence of embedded devices in our daily lives, AI at the edge has naturally attracted a great deal of interest. However, recent research has shown that real-world data often deviated from training data, resulting in poor predictive performance [40]. In such a context, continual learning proved to be a strong candidate for ensuring ongoing adaptation after deployment. Similar to the human learning process, it provides models with the ability to adapt to new incoming tasks, ideally without falling into the trap of catastrophic forgetting (i.e., degrading performance on older tasks in favor of new ones) [11, 33, 19]. In this respect, the very low memory and computing resources of extreme edge devices are an obstacle to the high cost of backpropagation. Lin *et al.* have shown that this challenge can be overcome by selectively fine-tuning a sub-graph of pre-trained networks on a general dataset [25]. Their work started a line of research showing the possibility of improvement through careful selection of channels to be updated at training time [23, 35]. However, none of these works addresses the computational cost of training a neuron, which is related to both the cost of loading the necessary weights and activations into RAM.

**Gradient Filtering.** In their work, Yang *et al.* demonstrate that it is possible to significantly reduce the memory and computational cost of full layer learning using a method called gradient filtering, which reduces the number of unique elements in the gradient map [53]. In such a frame and with the same specific objective, we propose to compress the activation maps by using tensor decomposition to minimize the memory cost. We perform this while providing a strong guarantee on the signal loss: unlike gradient filtering, we adaptively size the decomposed tensor to guarantee minimal information loss. In the next section, we will illustrate our proposed approach.

## 3    Method

In this section, we motivate our compression proposal by exposing the major memory bottleneck of backpropagation (Sec. 3.1). We then present our contribution which is the compression of activation maps through two well-known low-rank decomposition strategies, namely Singular Values Decomposition (SVD) and its tensor counterpart Higher Order SVD (HOSVD) (Sec. 3.2). In Sec. 3.3 we explore the induced changes to backpropagation calculation and in Sec. 3.4 we study the resulting memory and computational complexity, alongside an evaluation of the error introduced. Our ultimate goal here is to reduce the memory footprint of backpropagation (BP).

## 3.1 The Memory Bottleneck of Backpropagation

Following the formalism introduced in [2], we consider a simple convolutional neural network (CNN) represented as a sequence of $n$ convolutional layers (excluding the bias for simplicity):

$$\mathcal{F}(\mathcal{X}) = (\mathcal{C}_{\mathcal{W}_n} \circ \mathcal{C}_{\mathcal{W}_{n-1}} \circ \cdots \circ \mathcal{C}_{\mathcal{W}_2} \circ \mathcal{C}_{\mathcal{W}_1})(\mathcal{X}), \tag{1}$$

where $\mathcal{W}_i \in \mathbb{R}^{C' \times C \times D \times D}$ represents the filter parameters of the $i^{th}$ layer, with a kernel size of $D \times D$. This layer receives a $C$-channel input and produces an output with $C'$ channels. For this layer, we denote the input and output activation tensors as $\mathcal{A}_i \in \mathbb{R}^{B \times C \times H \times W}$ and $\mathcal{A}_{i+1} \in \mathbb{R}^{B \times C' \times H' \times W'}$, respectively. Note that $H$ and $W$ denote the height and width of each element of the input, while $H'$ and $W'$ denote the height and width of each element of the output, with $B$ as the minibatch size. The loss $\mathcal{L}$ is computed at the output of the network and backpropagated until the $i^{th}$ layer as $\frac{\partial \mathcal{L}}{\partial \mathcal{A}_{i+1}}$. At this stage, the filter parameters are updated thanks to the computation of $\frac{\partial \mathcal{L}}{\partial \mathcal{W}_i}$ and the loss is propagated to the previous layer as $\frac{\partial \mathcal{L}}{\partial \mathcal{A}_i}$. The computation of these terms follows the chain rule:

$$\frac{\partial \mathcal{L}}{\partial \mathcal{W}_i} = \frac{\partial \mathcal{L}}{\partial \mathcal{A}_{i+1}} \cdot \frac{\partial \mathcal{A}_{i+1}}{\partial \mathcal{W}_i} = \mathrm{conv}\left(\mathcal{A}_i, \frac{\partial \mathcal{L}}{\partial \mathcal{A}_{i+1}}\right), \tag{2}$$

$$\frac{\partial \mathcal{L}}{\partial \mathcal{A}_i} = \frac{\partial \mathcal{L}}{\partial \mathcal{A}_{i+1}} \cdot \frac{\partial \mathcal{A}_{i+1}}{\partial \mathcal{A}_i} = \mathrm{conv}_{\text{full}}\left[\frac{\partial \mathcal{L}}{\partial \mathcal{A}_{i+1}}, \mathrm{rot}(\mathcal{W}_i)\right], \tag{3}$$

where $\mathrm{conv}(\cdot)$ is the traditional convolution operation, convolving the kernel $\frac{\partial \mathcal{L}}{\partial \mathcal{A}_{i+1}}$ with the input $\mathcal{A}_i$; while $\mathrm{conv}_{\text{full}}(\cdot)$ is the convolutional operation that naturally maps the input $\frac{\partial \mathcal{L}}{\partial \mathcal{A}_{i+1}}$ to an output with the same dimensions as $\mathcal{A}_i$ by using the $180°$ rotated kernel $\mathcal{W}_i$.

From (2), it is clear that computing the weight derivatives requires to load input activation $\mathcal{A}_i$ and (3) shows that the weights $\mathcal{W}_i$ must be loaded into memory to calculate activation derivatives. We obtain the same conclusions for linear layers and provide the demonstration in Appendix A.2.

To save memory, two possibilities naturally emerge: either compressing the weights or compressing the activation. Weight compression is an extensively explored matter for network acceleration and we do not intend to further this area of research in this work. This leaves us with activation compression, which is still a new domain of exploration, but shows great potential for prospectively enabling on-device backpropagation. In such a regard, thanks to its strong theoretical grounding, tensor decomposition stands as a promising approach.

## 3.2 Tensor Decomposition

We will present here first the general Singular Value Decomposition (SVD) approach, which is then extended to a multidimensional variant (HOSVD), instrumental for our purposes.

**SVD.** Given a matrix $A \in \mathbb{R}^{M \times N}$, applying SVD to $A$ consists in a factorization of the form:

$$A = U\Sigma V^T, \qquad U \in \mathbb{R}^{M \times M}, \quad \Sigma \in \mathbb{R}^{M \times N}, \quad V \in \mathbb{R}^{N \times N}, \tag{4}$$

where $\Sigma$ is a rectangular diagonal matrix composed of $r$ singular values $s_{i \in [1,r]}$ with $r$ the rank of $A$. From this, we can deduce the amount of overall variance $\sigma_i^2$ explained by the $i^{th}$ pair of SVD vectors as $\sigma_i^2 = s_i^2 / \sum_j s_j^2$.

Let us assume the singular values in $\Sigma$ are ordered in descending order, $s_i \geq s_j, \forall i \leq j$. Given a desired threshold of cumulated explained variance $\varepsilon \in [0, 1]$, it becomes easy to find the "truncation threshold" which is the minimal $K \in [1, r]$ such that $\sum_{i=1}^{K} \sigma_i^2 \geq \varepsilon$. We can then approximate $A$ by only selecting the $K$ first columns of $U$ and $V$ and the $K$ first singular values from $\Sigma$, according to:

$$\tilde{A} = U_{(K)}\Sigma_{(K)}V_{(K)}^T, \qquad U_{(K)} \in \mathbb{R}^{M \times K}, \quad \Sigma_{(K)} \in \mathbb{R}^{K \times K}, \quad V_{(K)} \in \mathbb{R}^{N \times K}. \tag{5}$$

Historically, SVD was the first example of tensor decomposition applied to neural network compression and acceleration [55] but applied to the model's parameters [49]. We take the SVD decomposition to the activation maps as one possible baseline to compare with more complex low-rank compression solutions.

In the case of convolutional layer $i$, since SVD is designed for matrix decomposition, we will simply reshape the activation $\mathcal{A}_i$ as matrix $A_i$ of dimensions $B \times (CHW)$. Given a desired level of explained variance $\varepsilon$, we obtain the decomposition described in (5). We then store in memory the terms

$U_{(K)} \times \Sigma_{(K)}$ and $V_{(K)}^T$, meaning that instead of storing $\Theta_{\text{space}}(BCHW)$ unique elements in memory, we are only storing $\Theta_{\text{space}}[K(B + CHW)]$ unique elements. Regarding linear layers, activations are $M \times N$ matrices, and applying SVD is much more straightforward, leading to the storage cost of $\Theta_{\text{space}}[K(M + N)]$ instead of $\Theta_{\text{space}}(MN)$ elements. The larger the explained variance, the closer $\tilde{A}$ will be to $A$, intuitively allowing for better estimation when performing backpropagation. However, this also means a larger $K$ which results in a larger memory occupation. The goal is then to find an efficient trade-off between the desired explained variance and compression rates.

**HOSVD.** By construction, SVD is designed for matrix decomposition. It was demonstrated that the reshaping operation on tensors introduced structure information distortion, leading to sub-optimal performance [49]. Other methods more suited for tensor decomposition such as Canonical-Polyadic (CP) decomposition [20] or Tucker decomposition [43] were naturally introduced to tackle this issue. Given a $n^{th}$-order tensor $\mathcal{T} \in \mathbb{R}^{M_1 \times M_2 \times \cdots \times M_n}$, its Tucker decomposition corresponds to:

$$\mathcal{T} = \mathcal{S} \times_1 U^{(1)} \times_2 U^{(2)} \times_3 \cdots \times_n U^{(n)}, \tag{6}$$

where $\mathcal{S} \in \mathbb{R}^{L_1 \times L_2 \times \cdots \times L_n}$ is the core tensor which can be viewed as a compressed version of $\mathcal{T}$, $U^{(j)} \in \mathbb{R}^{M_j \times L_j}$ are the factor matrices and their columns correspond to the principal components over the $j^{th}$ mode. The $i$-mode product "$\times_i$" of a $n^{th}$-order tensor $\mathcal{G} \in \mathbb{R}^{P_1 \times P_2 \times \cdots \times P_n}$ and a matrix $B \in \mathbb{R}^{Q \times P_i}$ is a $n^{th}$-order tensor $\mathcal{R} \in \mathbb{R}^{P_1 \times \cdots \times P_{i-1} \times Q \times P_{i+1} \times \cdots \times P_n}$ which can be expressed as:

$$\mathcal{R}_{p_1, \ldots, p_{i-1}, q, p_{i+1}, \ldots, p_n} = \mathcal{G} \times_i Q = \sum_{p_i=1}^{P_i} g_{p_1, p_2, \ldots, p_n} b_{q, p_i}. \tag{7}$$

In such a setup, Higher-Order SVD is a specific case of Tucker decomposition where the factor matrices $U^{(j)}$ are orthogonal [44]. As a follow-up to SVD, we propose to compress activation tensors through HOSVD, with the intuition that each dimension encodes a different variance, potentially providing enhanced compression rates for equivalent performance and vice-versa. Similarly to traditional SVD, we can truncate $\mathcal{S}$ and each $U^{(j)}$ along each mode given a desired level of explained variance resulting in:

$$\tilde{\mathcal{T}} = \hat{\mathcal{S}} \times_1 U^{(1)}_{(K_1)} \times_2 \cdots \times_n U^{(n)}_{(K_n)} \approx \mathcal{T}, \tag{8}$$

where $U^{(j)}_{(K_j)} \in \mathbb{R}^{M_j \times K_j}$ corresponds to the $K_j$ first columns of $U^{(j)}$ and $\hat{\mathcal{S}} \in \mathbb{R}^{K_1 \times \cdots \times K_n}$ is the truncated version of $\mathcal{S}$. In the following section, we focus on HOSVD decomposition, more precisely on the alterations induced to the backpropagation graph, the resulting compression and speedup ratio, and the error bound. An equivalent analysis can be made for SVD.

### 3.3 Backpropagation with Compressed Activations

Prior works [13, 18] already demonstrated the possibility of performing backpropagation operations in the decomposed space without relying on the recomposition of compressed tensors. In their work, Kim *et al.* compress a $4^{th}$-order kernel tensor $\mathcal{W} \in \mathbb{R}^{C' \times C \times D \times D}$ through Tucker decomposition limited to mode 1 and 2:

$$\mathcal{W} = \mathcal{W}' \times_1 U^{(1)} \times_2 U^{(2)}, \qquad \mathcal{W}' \in \mathbb{R}^{L_1 \times L_2 \times D \times D}, \quad U^{(1)} \in \mathbb{R}^{C' \times L_1}, \quad U^{(2)} \in \mathbb{R}^{C \times L_2}. \tag{9}$$

Then, it is shown that the output is calculated through

$$\mathcal{Y} = \text{conv}_{1 \times 1}\{U^{(2)}, \text{conv}_{D \times D}[\mathcal{W}', \text{conv}_{1 \times 1}(U^{(1)}, \mathcal{X})]\}. \tag{10}$$

Our problem is similar although with some fundamental differences linked to the need to reconstruct *gradients*. A convolution operator is still required to compute the weight derivatives as introduced in (2): we apply a specific case of Tucker to one of the two components, namely the activation $\mathcal{A}_i$. Following the same reasoning, we derive that the computation of $\frac{\partial \mathcal{L}}{\partial \mathcal{W}_i}$ in (2) becomes:

$$\frac{\partial \mathcal{L}}{\partial \mathcal{W}_i} = \text{conv}_{1 \times 1}\left\{\text{conv}_*\left[\text{conv}_{1 \times 1}\left(\text{conv}_{1 \times 1}\left(\hat{\mathcal{S}}, \underline{U}^{(3)}_{(K_3)}\right), \underline{U}^{(4)}_{(K_4)}\right), \text{conv}_{1 \times 1}\left(\frac{\partial \mathcal{L}}{\partial \mathcal{A}_{i+1}}, U^{(1)}_{(K_1)}\right)\right], U^{(2)}_{(K_2)}\right\}, \tag{11}$$

where $\text{conv}_*$ is a 2D convolution with a specific kernel size as defined in (23), and $\underline{U}^{(j)}_{(K_j)}$ is the vertically padded version of $U^{(j)}_{(K_j)}$ as defined in (19). Demonstration details are available in

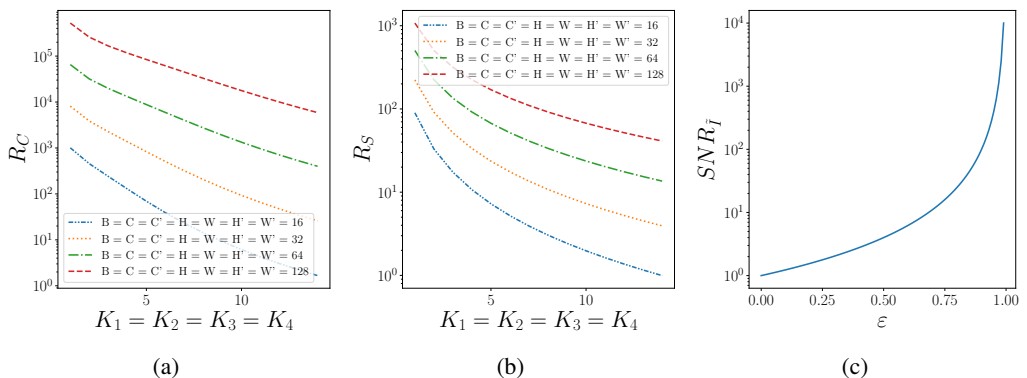

Figure 2: For a single convolutional layer with minibatch size $B$, **(a)** and **(b)** illustrate the predicted changes in compression rate $R_C$ and speedup ratios $R_S$ as functions of $K_j$, when comparing HOSVD with vanilla training, respectively. **(c)** shows the evolution of the SNR with retained variance $\varepsilon$.

Appendix A.3. This way, we can compute the approximated weight derivative without reconstructing the activation, through the successive computation of simpler convolutions.

Fig. 1 illustrates our method. During training, the forward pass proceeds as usual, with one key modification to memory management. Instead of keeping complete activation maps in memory, we store only their principal components, which are derived from the decomposition process. During the backward pass, the principal components are retrieved from memory and used for calculations as described in (11).

### 3.4 Complexity and Error Analyis

**Computational Speedup and Space Complexity.** For simplicity we assume here that $\mathcal{A}_{i+1}$ and $\mathcal{A}_i$ have the same shape and that the truncation thresholds across all modes are equal for HOSVD, i.e., $K_1 = K_2 = K_3 = K_4$. Considering different thresholding values, we can predict the relative improvement of HOSVD to vanilla training in both space complexity (30) and computational speedup (29), for a backward pass. As illustrated in Fig. 2a and Fig. 2b, our method is more effective for both space complexity and latency for the backward pass when the truncation threshold decreases and the activation size increases. More details are provided in Appendix A.4.
We hypothesize that the first components along each dimension are enough to encode most of the variance, implying that with relatively low values of $K_j$, we can achieve good training performance. We later empirically confirm this assumption in a variety of experimental setups (Sec. 4.2).

**Signal to Noise Ratio.** For simplicity, we rewrite in this paragraph $\frac{\partial \mathcal{L}}{\partial \mathcal{W}_i}$ as $\Delta \mathcal{W}$, $\mathcal{A}_i$ as $\mathcal{I}$ and $\frac{\partial \mathcal{L}}{\partial \mathcal{A}_{i+1}}$ as $\Delta \mathcal{Y}$. Regarding the error introduced by truncating $\mathcal{I}$ in order to retain the components corresponding to a given level of explained variance $\varepsilon \in [0, 1]$, we demonstrate that the energy contained in the resulting gradient is equal to the energy contained in $\tilde{\mathcal{I}}$. We note $I$, $\Delta W$ and $\Delta Y$ as the input activation, weight derivative, and output activation derivative in the frequency domain; $I[u, v]$, the spectrum value at frequency $(u, v)$. Applying the discrete Fourier transformation to $\tilde{\mathcal{I}}$ gives us $\tilde{I} = \varepsilon I$, which we use to compute the signal to noise ratio $SNR_{\tilde{I}}$:

$$SNR_{\tilde{I}} = \frac{\sum_{(u,v)} I[u, v]^2}{\sum (I[u, v] - \varepsilon I[u, v])^2} = \frac{1}{(1 - \varepsilon)^2}. \tag{12}$$

Similarly, in the frequency domain, as the convolutional operation becomes a regular multiplication, $\Delta \tilde{\mathcal{W}}$ becomes $\Delta \tilde{W} = \tilde{I} \Delta Y = \varepsilon I \Delta Y$. As for (12), we obtain $SNR_{\Delta \tilde{W}} = (1 - \varepsilon)^{-2} = SNR_{\tilde{I}}$. A visual representation is provided in Fig. 2c. Analytically, this confirms the idea that the closer $\varepsilon$ is to 1, the larger the energy is transferred from the compressed input to the weight derivatives (for $\varepsilon = 0.8$ we have $SNR_{\Delta \tilde{W}} = 25$).
Additionally, in our setup since we are only compressing the activations and given (2) and (3), the introduced error only transfers to the weight derivative for each layer. The activation derivatives that are propagated from one layer to another are exact computations as they do not involve the

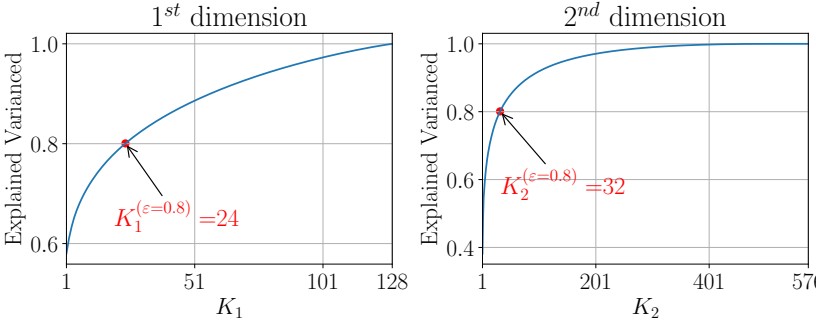

Figure 3: Explained variance $\varepsilon$ for the first two dimensions of the activation map in the $4^{th}$ last layer when fine-tuning the last four layers of MCUNet using HOSVD on CIFAR-10, following setup A.

activations. This means that the error introduced when compressing the activations at each layer does not accumulate through the network.

# 4 Experiments

In this section, we describe the experiments conducted to support the claims presented in Sec. 3. First, we introduce the setups used for our experiments (Sec. 4.1); then, we analyze the energy distribution in the different dimensions of HOSVD, providing an overview of the typical values of $K$ (Sec. 4.2); finally, we test our algorithm in different setups, state-of-the-art architectures and datasets to evaluate the tradeoff between accuracy and memory footprint (Sec. 4.3). Experiments were performed using a NVIDIA RTX 3090Ti and the source code uses PyTorch 1.13.1.

## 4.1 Experimental setup

To validate our approach, we perform a variety of computer vision experiments split across two tasks, classification and segmentation.
**Classification.** We explore two types of fine-tuning policies:

- *Full fine-tuning* (referred to as "setup A"): Following conventional fine-tuning trends, we load models pre-trained on ImageNet [21] and we fine-tune them on a variety of downstream datasets (CIFAR-10, CIFAR-100, CUB [46], Flowers [32] and Pets [54]).

- *Half fine-tuning* (referred to as "setup B"): Following Yang *et al.* approach, each classification dataset (ImageNet, CIFAR-10/100) is split into two non-i.i.d. partitions of equal size using the FedAvg [29] method. The partitions are then split as follows: $80\%$ for training and $20\%$ for validation. The first partition is used for pretraining, and the second partition is used for finetuning.

**Semantic Segmentation.** Similarly to the half fine-tuning method, we reproduce Yang *et al.* segmentation setup: we fine-tune models pretrained on Cityscapes [6] by MMSegmentation [5]. Here there is only one downstream dataset which is Pascal-VOC12 [3]. Experimental details for both tasks (hyper-parameters, policy, etc.) are provided in Appendix B.2.

**Memory Logging.** For HOSVD and SVD, instead of focusing on compressing the tensor based on rank, we control compression through the desired amount of retained information. As a consequence, we cannot explicitly control the memory usage of the principal components. In the results presented below, we will always include two columns displaying peak and average memory, along with their standard deviations.

## 4.2 Explained Variance Evolution

In this section, we conduct experiments to fine-tune the last four convolutional layers of MCUNet [24] following Setup A, and CIFAR-10 as the downstream dataset. Using HOSVD with $\varepsilon$ set to 0.8, Fig. 3 shows the explained variance retained across dimension $j$ as a function of $K_j$, where $j = \{1, 2\}$ corresponds to the two largest dimensions of the activation map. We define $K_j^{(\varepsilon=x)}$ as the number of principal components required to retain at least a fraction $x$ of the explained variance.

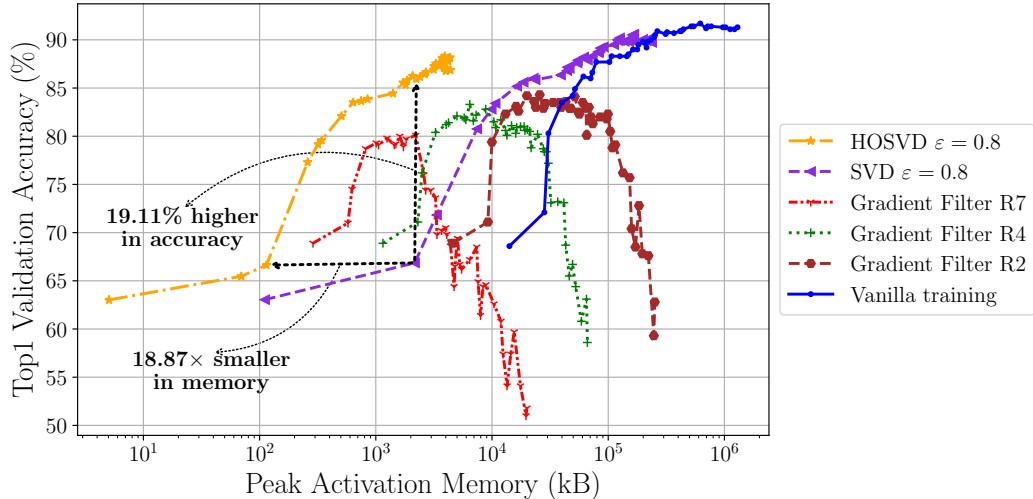

Figure 5: Performance curves of an MCUNet pre-trained on ImageNet and finetuned on CIFAR-10 with different activation compression strategies.

We observe that along the largest dimensions (batch size and number of channels), less than 20% of the components capture more than 80% of the explained variance, confirming the assumption proposed in Sec. 3.4. Additionally, the curves observed present a logarithmic behavior, hinting at the possibility of reaching high explained variance with little loss regarding the compression ratio. This is especially important as the $SNR$ transmitted to the weight derivatives increases quadratically to the explained variance (Fig. 2c). The results for the other layers are presented in Appendix B.3.

Fig. 4 illustrates the results of performing HOSVD with different explained variance thresholds $\varepsilon$. The results are averaged over three different random seeds. For $\varepsilon$ smaller than 0.8, we observe that as it increases, we achieve sig-

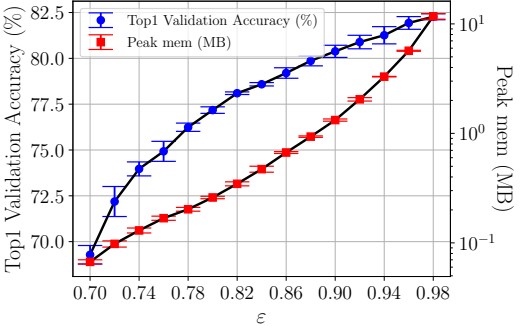

Figure 4: Behavior of top1 validation accuracy and peak memory when applying HOSVD with different explained variance thresholds $\varepsilon$ when finetuning the last four convolutional layers of an MCUNet model using the CIFAR-10 dataset on setup A.

nificant gains in accuracy, along with impressive compression ratios. However, when $\varepsilon$ exceeds 0.8, the accuracy growth starts slowing down. Above the 0.9 threshold, the exponential growth of peak memory results in a worsened tradeoff between accuracy and compression ratio. Therefore, in subsequent experiments presented in this paper, we will use $\varepsilon$ values of 0.8 and 0.9.

## 4.3 Main Results

**MCUNet on classification with setup A.** In this experiment, we fine-tune on CIFAR-10 an MCUNet pre-trained on ImageNet, with the number of finetuned layers increasing from 1 to 42 (all layers). We compare vanilla training, and gradient filtering with patch sizes of 2, 4, and 7, SVD, and HOSVD with an explained variance threshold of 0.8. For each method, we compare the effect of fine-tuning different model depths. Fig. 5 presents the performance curves for our experiments, with the X-axis representing activation memory in kiloBytes (kB) on a logarithmic scale and the Y-axis representing the highest validation accuracy. Each marker indicates the number of convolutional layers finetuned. For example, on the yellow curve representing the HOSVD method, the marker closest to the Y-axis shows the result when finetuning the last convolutional layer, the next marker represents finetuning the last two convolutional layers, and so on. The most effective method is the one whose performance curve trends towards the upper-left corner of the plot.

Table 1: Experimental results on ImageNet-1k. "#Layers" refers to the number of fine-tuned convolutional layers (counted from the end of the model). Activation memory consumption is shown in MegaBytes (MB).

| Method | MobileNetV2 | | | | Method | ResNet18 | | | |
|---|---|---|---|---|---|---|---|---|---|
| | #Layers | Acc ↑ | Peak Mem (MB) ↓ | Mean Mem (MB) ↓ | | #Layers | Acc ↑ | Peak Mem (MB) ↓ | Mean Mem (MB) ↓ |
| Vanilla training | All | 74.0 | 1651.84 | $1651.84 \pm 0.00$ | Vanilla training | All | 72.8 | 532.88 | $532.88 \pm 0.00$ |
| | 2 | 62.6 | 15.31 | $15.31 \pm 0.00$ | | 2 | 69.9 | 12.25 | $12.25 \pm 0.00$ |
| | 4 | 65.8 | 28.71 | $28.71 \pm 0.00$ | | 4 | 71.5 | 30.63 | $30.63 \pm 0.00$ |
| Gradient Filter R2 | 2 | 62.6 | 5.00 | $5.00 \pm 0.00$ | Gradient Filter R2 | 2 | 68.7 | 4.00 | $4.00 \pm 0.00$ |
| | 4 | 65.2 | 9.38 | $9.38 \pm 0.00$ | | 4 | 69.3 | 7.00 | $7.00 \pm 0.00$ |
| SVD ($\varepsilon = 0.8$) | 2 | 61.7 | 4.97 | $4.92 \pm 0.08$ | SVD ($\varepsilon = 0.8$) | 2 | 69.5 | 7.88 | $7.71 \pm 0.21$ |
| | 4 | 65.2 | 14.76 | $14.61 \pm 0.09$ | | 4 | 71.1 | 19.98 | $19.72 \pm 0.28$ |
| SVD ($\varepsilon = 0.9$) | 2 | 62.3 | 8.97 | $8.91 \pm 0.08$ | SVD ($\varepsilon = 0.9$) | 2 | 69.7 | 9.86 | $9.77 \pm 0.13$ |
| | 4 | 65.5 | 20.35 | $20.20 \pm 0.07$ | | 4 | 71.3 | 24.81 | $24.66 \pm 0.17$ |
| HOSVD ($\varepsilon = 0.8$) | 2 | 61.1 | 0.15 | $0.15 \pm 0.00$ | HOSVD ($\varepsilon = 0.8$) | 2 | 69.2 | 0.97 | $0.91 \pm 0.05$ |
| | 4 | 63.9 | 0.73 | $0.68 \pm 0.03$ | | 4 | 70.5 | 2.89 | $2.74 \pm 0.12$ |
| HOSVD ($\varepsilon = 0.9$) | 2 | 61.8 | 0.43 | $0.43 \pm 0.01$ | HOSVD ($\varepsilon = 0.9$) | 2 | 69.5 | 2.73 | $2.63 \pm 0.10$ |
| | 4 | 64.8 | 1.92 | $1.76 \pm 0.08$ | | 4 | 71.1 | 7.96 | $7.66 \pm 0.21$ |

| Method | MCUNet | | | | Method | ResNet34 | | | |
|---|---|---|---|---|---|---|---|---|---|
| | #Layers | Acc ↑ | Peak Mem (MB) ↓ | Mean Mem (MB) ↓ | | #Layers | Acc ↑ | Peak Mem (MB) ↓ | Mean Mem (MB) ↓ |
| Vanilla training | All | 67.4 | 632.98 | $632.98 \pm 0.00$ | Vanilla training | All | 75.6 | 839.04 | $839.04 \pm 0.00$ |
| | 2 | 62.1 | 13.78 | $13.78 \pm 0.00$ | | 2 | 69.6 | 12.25 | $12.25 \pm 0.00$ |
| | 4 | 64.7 | 19.52 | $19.52 \pm 0.00$ | | 4 | 72.2 | 24.50 | $24.50 \pm 0.00$ |
| Gradient Filter R2 | 2 | 61.8 | 4.50 | $4.50 \pm 0.00$ | Gradient Filter R2 | 2 | 68.8 | 4.00 | $4.00 \pm 0.00$ |
| | 4 | 64.4 | 6.38 | $6.38 \pm 0.00$ | | 4 | 70.9 | 8.00 | $8.00 \pm 0.00$ |
| SVD ($\varepsilon = 0.8$) | 2 | 62.0 | 7.62 | $7.51 \pm 0.12$ | SVD ($\varepsilon = 0.8$) | 2 | 69.2 | 6.70 | $6.49 \pm 0.29$ |
| | 4 | 64.5 | 10.59 | $10.37 \pm 0.20$ | | 4 | 71.8 | 14.68 | $14.24 \pm 0.50$ |
| SVD ($\varepsilon = 0.9$) | 2 | 62.1 | 10.32 | $10.26 \pm 0.08$ | SVD ($\varepsilon = 0.9$) | 2 | 69.4 | 9.10 | $8.96 \pm 0.23$ |
| | 4 | 64.6 | 14.39 | $14.26 \pm 0.13$ | | 4 | 72.0 | 19.11 | $18.83 \pm 0.37$ |
| HOSVD ($\varepsilon = 0.8$) | 2 | 61.7 | 0.48 | $0.43 \pm 0.04$ | HOSVD ($\varepsilon = 0.8$) | 2 | 68.7 | 0.30 | $0.27 \pm 0.02$ |
| | 4 | 63.9 | 0.88 | $0.78 \pm 0.07$ | | 4 | 71.1 | 1.11 | $1.02 \pm 0.07$ |
| HOSVD ($\varepsilon = 0.9$) | 2 | 62.0 | 1.32 | $1.27 \pm 0.06$ | HOSVD ($\varepsilon = 0.9$) | 2 | 69.2 | 0.71 | $0.65 \pm 0.05$ |
| | 4 | 64.4 | 2.52 | $2.36 \pm 0.15$ | | 4 | 71.9 | 3.24 | $3.09 \pm 0.13$ |

We observe that as the number of finetuned layers increases, the gradient filtering accuracy increases up to a certain point, and then deteriorates, whereas the accuracy for SVD, HOSVD, and vanilla training keeps improving with additional layers, along a similar trend. Intuitively, gradient filtering might propagate errors through the layers during training, while our HOSVD method keeps the error introduced on each individual layer confined to that specific layer as demonstrated in Sec. 3.4. Moreover, we observe that for identical depths, SVD accuracies consistently exceed HOSVD ones. We hypothesize that this is due to HOSVD performing SVD across all modes of the tensor: it potentially loses information in all modes, whereas SVD only loses information in one mode.
Compared to SVD, HOSVD significantly reduces memory, given equivalent accuracy levels (up to $18.87$ times). Similarly, for equivalent memory usage, HOSVD yields greatly improved accuracy compared to SVD (up to $19.11\%$). When compared to methods such as gradient filtering and vanilla training, given an equivalent memory budget, HOSVD yields significantly higher accuracies. Notably, fine-tuning all layers with HOSVD requires much less memory than fine-tuning only the last layer with vanilla training, which is consistent with the theoretical compression ratio shown in Fig. 2a. Additional experiments with setup A can be found in Appendix B.5.

**ImageNet Classification with setup B.** Table 1 presents the classification performance and memory consumption for MobileNetV2 [37], ResNet18, and ResNet34 [15] models using various methods, including vanilla training, gradient filtering, HOSVD, and SVD on the ImageNet dataset. We observe in most cases that, for the same depth, SVD and HOSVD are competitive with the gradient filtering method in terms of performance while reaching much lower activation memory with HOSVD.

**Segmentation.** Table 2 reports the segmentation performance and memory consumption for a variety of architectures as presented in Yang *et al.*. In general, we observe that increasing the level of explained variance from $0.8$ to $0.9$ substantially increases the performance with little trade-off on retained activation memory. This further confirms that most of the explained variance is contained in the first few components, allowing for efficient generalization with high compression rates.

## 5    Conclusion

In this work, we have addressed one of the main obstacles to on-device training. Inspired by traditional low-rank optimization approaches, we propose to compress activation maps by tensor decomposition, using HOSVD as a supporting example (Sec. 3.2). We demonstrate that the compression error introduced is bounded and confined to each individual layer considered, validating our approach

Table 2: Experimental results for semantic segmentation. mIoU is the mean Intersection over Union, and mAcc is the micro averaged accuracy.

| Method | #Layers | mIoU ↑ | mAcc ↑ | Peak Mem (MB) ↓ | Mean Mem (MB) ↓ | Method | #Layers | mIoU ↑ | mAcc ↑ | Peak Mem (MB) ↓ | Mean Mem (MB) ↓ |
|---|---|---|---|---|---|---|---|---|---|---|---|
| | | | **PSPNet [57]** | | | | | | **PSPNet-M [57]** | | |
| Vanilla training | All | 54.97 | 68.46 | 920.78 | 920.78 ± 0.00 | Vanilla training | All | 48.92 | 62.11 | 2622.49 | 2622.49 ± 0.00 |
| | 5 | 39.36 | 51.79 | 128.00 | 128.00 ± 0.00 | | 5 | 36.22 | 46.31 | 104.00 | 104.00 ± 0.00 |
| | 10 | 53.17 | 67.18 | 352.00 | 352.00 ± 0.00 | | 10 | 45.62 | 58.35 | 604.00 | 604.00 ± 0.00 |
| Gradient Filter | 5 | 39.34 | 51.59 | 8.00 | 8.00 ± 0.00 | Gradient Filter | 5 | 35.73 | 45.78 | 6.50 | 6.50 ± 0.00 |
| | 10 | 51.2 | 65.01 | 22.00 | 22.00 ± 0.00 | | 10 | 44.89 | 57.38 | 37.75 | 37.75 ± 0.00 |
| SVD ($\varepsilon = 0.8$) | 5 | 38.97 | 50.04 | 90.40 | 85.64 ± 2.80 | SVD ($\varepsilon = 0.8$) | 5 | 34.98 | 44.48 | 58.50 | 57.17 ± 0.77 |
| | 10 | 52.03 | 65.44 | 238.20 | 232.50 ± 4.71 | | 10 | 44.73 | 56.83 | 377.68 | 369.24 ± 6.93 |
| SVD ($\varepsilon = 0.9$) | 5 | 39.34 | 50.92 | 111.20 | 108.88 ± 1.43 | SVD ($\varepsilon = 0.9$) | 5 | 35.51 | 45.51 | 78.65 | 78.65 ± 0.00 |
| | 10 | 52.7 | 66.39 | 295.60 | 292.46 ± 2.50 | | 10 | 45.79 | 58.97 | 482.00 | 475.17 ± 4.79 |
| HOSVD ($\varepsilon = 0.8$) | 5 | 38.11 | 49.29 | 0.47 | 0.32 ± 0.08 | HOSVD ($\varepsilon = 0.8$) | 5 | 33.40 | 42.50 | 0.03 | 0.03 ± 0.00 |
| | 10 | 49.23 | 62.39 | 1.40 | 1.24 ± 0.08 | | 10 | 40.06 | 51.79 | 1.47 | 1.30 ± 0.06 |
| HOSVD ($\varepsilon = 0.9$) | 5 | 39.03 | 50.29 | 2.26 | 1.70 ± 0.30 | HOSVD ($\varepsilon = 0.9$) | 5 | 34.09 | 43.69 | 0.07 | 0.07 ± 0.00 |
| | 10 | 52.11 | 65.5 | 8.18 | 6.44 ± 0.49 | | 10 | 44 | 56.9 | 8.20 | 6.16 ± 0.62 |
| | | | **DLV3 [3]** | | | | | | **DLV3-M [3]** | | |
| Vanilla training | All | 58.44 | 72.03 | 1128.02 | 1128.02 ± 0.00 | Vanilla training | All | 55.87 | 69.47 | 2758.01 | 2758.01 ± 0.00 |
| | 5 | 40.75 | 52.95 | 336.00 | 336.00 ± 0.00 | | 5 | 38.38 | 49.61 | 240.00 | 240.00 ± 0.00 |
| | 10 | 55.04 | 69.07 | 560.00 | 560.00 ± 0.00 | | 10 | 47.91 | 61.67 | 620.00 | 620.00 ± 0.00 |
| Gradient Filter | 5 | 32.18 | 42.93 | 27.47 | 27.47 ± 0.00 | Gradient Filter | 5 | 35.7 | 46.71 | 20.71 | 20.71 ± 0.00 |
| | 10 | 47.44 | 60.08 | 83.47 | 83.47 ± 0.00 | | 10 | 45.4 | 58.97 | 65.62 | 65.62 ± 0.00 |
| SVD ($\varepsilon = 0.8$) | 5 | 39.75 | 51.69 | 242.30 | 240.91 ± 0.77 | SVD ($\varepsilon = 0.8$) | 5 | 36.78 | 47.15 | 169.00 | 166.87 ± 1.26 |
| | 10 | 52.69 | 66.2 | 381.00 | 376.81 ± 3.02 | | 10 | 46.22 | 57.82 | 374.13 | 366.76 ± 5.66 |
| SVD ($\varepsilon = 0.9$) | 5 | 40.47 | 52.09 | 291.40 | 291.40 ± 0.00 | SVD ($\varepsilon = 0.9$) | 5 | 37.59 | 48.24 | 204.50 | 200.69 ± 1.80 |
| | 10 | 53.86 | 67.61 | 472.70 | 466.52 ± 4.71 | | 10 | 47.4 | 59.4 | 493.88 | 485.56 ± 5.77 |
| HOSVD ($\varepsilon = 0.8$) | 5 | 38.52 | 50.14 | 2.66 | 2.62 ± 0.01 | HOSVD ($\varepsilon = 0.8$) | 5 | 35.55 | 45.64 | 0.63 | 0.55 ± 0.03 |
| | 10 | 50.11 | 63.14 | 1.93 | 1.48 ± 0.14 | | 10 | 42.68 | 54.17 | 1.04 | 0.96 ± 0.05 |
| HOSVD ($\varepsilon = 0.9$) | 5 | 40.19 | 52.3 | 12.64 | 12.35 ± 0.10 | HOSVD ($\varepsilon = 0.9$) | 5 | 37.06 | 47.73 | 4.56 | 4.16 ± 0.15 |
| | 10 | 52.26 | 65.8 | 9.23 | 7.24 ± 0.63 | | 10 | 45.75 | 57.61 | 5.52 | 4.79 ± 0.29 |
| | | | **FCN [27]** | | | | | | **UPerNet [48]** | | |
| Vanilla training | All | 45.36 | 59.53 | 952.00 | 952.00 ± 0.00 | Vanilla training | All | 64.71 | 77.32 | 2168.78 | 2168.78 ± 0.00 |
| | 5 | 27.31 | 38.21 | 288.00 | 288.00 ± 0.00 | | 5 | 48.05 | 61.66 | 1380.00 | 1380.00 ± 0.00 |
| | 10 | 43.54 | 57.96 | 480.00 | 480.00 ± 0.00 | | 10 | 48.9 | 63.1 | 1436.00 | 1436.00 ± 0.00 |
| Gradient Filter | 5 | 27.24 | 38.1 | 18.00 | 18.00 ± 0.00 | Gradient Filter | 5 | 46.79 | 60.5 | 33.00 | 33.00 ± 0.00 |
| | 10 | 36.91 | 50.14 | 120.00 | 120.00 ± 0.00 | | 10 | 47.89 | 62.44 | 36.50 | 36.50 ± 0.00 |
| SVD ($\varepsilon = 0.8$) | 5 | 28.62 | 38.42 | 196.30 | 191.41 ± 2.12 | SVD ($\varepsilon = 0.8$) | 5 | 46.56 | 59.46 | 804.55 | 784.80 ± 17.28 |
| | 10 | 34.60 | 45.71 | 316.70 | 288.37 ± 13.78 | | 10 | 47.70 | 61.02 | 840.70 | 829.13 ± 5.39 |
| SVD ($\varepsilon = 0.9$) | 5 | 31.92 | 42.35 | 251.50 | 245.40 ± 1.74 | SVD ($\varepsilon = 0.9$) | 5 | 47.22 | 59.77 | 1078.38 | 1056.52 ± 14.42 |
| | 10 | 42.04 | 54.53 | 406.20 | 396.14 ± 4.89 | | 10 | 48.3 | 61.16 | 1126.50 | 1113.98 ± 11.49 |
| HOSVD ($\varepsilon = 0.8$) | 5 | 26.34 | 36.14 | 1.43 | 1.26 ± 0.08 | HOSVD ($\varepsilon = 0.8$) | 5 | 45.28 | 57.80 | 1.35 | 1.32 ± 0.02 |
| | 10 | 30.91 | 41.19 | 3.77 | 2.54 ± 0.72 | | 10 | 46.44 | 58.93 | 1.68 | 1.63 ± 0.03 |
| HOSVD ($\varepsilon = 0.9$) | 5 | 30.46 | 41.12 | 8.10 | 6.99 ± 0.38 | HOSVD ($\varepsilon = 0.9$) | 5 | 46.8 | 59.29 | 4.72 | 4.59 ± 0.10 |
| | 10 | 39.89 | 52.11 | 17.57 | 14.28 ± 1.55 | | 10 | 47.94 | 60.7 | 7.81 | 7.51 ± 0.12 |

through extensive experimentations. This work paves the way for a new line of research combining the accumulated knowledge on tensor decomposition strategies and the recent field of on-device learning.

## Acknowledgements

Part of this work was funded by Hi!PARIS Center on Data Analytics and Artificial Intelligence, by the European Union's HORIZON Research and Innovation Programme under grant agreement No 101120657, project ENFIELD (European Lighthouse to Manifest Trustworthy and Green AI) and by French National Research Agency (ANR-22-PEFT-0003 and ANR-22-PEFT-0007) as part of France 2030, the NF-NAI project and NF-FITNESS project.

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

# A  Additional Theoretical Details

## A.1  Forward pass Vs. Backward pass

We consider the same notations introduced in Sec. 3.1. Regarding the training of a convolutional layer:

- There is one convolution operation in each forward pass with a computational complexity of $\Theta_{\text{FLOPS}}(D^2 CC' BH'W')$

- In each backward pass, there are two convolution operations, corresponding to formulas (2) and (3), and one weight update operation, with computational complexities of $\Theta_{\text{FLOPS}}(D^2 CC' BH'W')$, $\Theta_{\text{FLOPS}}(D^2 CC' BHW)$ and $\Theta_{\text{FLOPS}}(D^2 CC')$, respectively.

From this, we can calculate the ratio of computational complexity between forward and backward pass $R_{\text{FLOPs}}$ as follows:

$$R_{\text{FLOPs}} = \frac{D^2 CC' BH'W'}{D^2 CC' BH'W' + D^2 CC' BHW + D^2 CC'}. \tag{13}$$

It is evident that $R_{\text{FLOPs}}$ is always smaller than one, indicating that the backward pass is always more computationally intensive than the forward pass.

## A.2  Backpropagation Derivatives in Linear Layers

With the same notations introduced in Sec. 3.1 but in the case of a linear neural network represented as a sequence of $n$ linear layers:

$$\mathcal{F}(X) = (\mathcal{F}_{W_n} \circ \mathcal{F}_{W_{n-1}} \circ \cdots \circ \mathcal{F}_{W_2} \circ \mathcal{F}_{W_1})(X), \tag{14}$$

where $W_i \in \mathbb{R}^{O \times I}$ corresponds to parameters matrix of the $i^{th}$ layer. This layer receives an $I$-feature input $A_i \in \mathbb{R}^{B \times I}$ with minibatch size $B$ to produce an $O$-feature output $A_{i+1} \in \mathbb{R}^{B \times O}$. The derivatives in that setup become:

$$\frac{\partial \mathcal{L}}{\partial W_i} = \frac{\partial \mathcal{L}}{\partial A_{i+1}} \cdot \frac{\partial A_{i+1}}{\partial W_i} = A_i^T \cdot \frac{\partial \mathcal{L}}{\partial A_{i+1}}, \tag{15}$$

$$\frac{\partial \mathcal{L}}{\partial A_i} = \frac{\partial \mathcal{L}}{\partial A_{i+1}} \cdot \frac{\partial A_{i+1}}{\partial A_i} = W_i^T \cdot \frac{\partial \mathcal{L}}{\partial A_{i+1}}. \tag{16}$$

The same conclusions as for convolutional layers can naturally be derived from these equations for linear layers, namely that activations and weights occupy most of the space in memory when performing backpropagation.

## A.3  Details of Backpropagation with Decomposed Activation Tensors

Since the convolution operation involves other variables such as stride, dilation, and groups, we provide a more precise description by rewriting in (2), $\frac{\partial \mathcal{L}}{\partial \mathcal{W}_i}$ as $\Delta \mathcal{W} \in \mathbb{R}^{N_{out} \times C \times D \times D}$, $\mathcal{A}_i$ as $\mathcal{I} \in \mathbb{R}^{B \times C \times H \times W}$ and $\frac{\partial \mathcal{L}}{\partial \mathcal{A}_{i+1}}$ as $\Delta \mathcal{Y} \in \mathbb{R}^{B \times C' \times H' \times W'}$ which gives us:

$$\Delta\mathcal{W}_{c'_g,c,k,l} = \sum_{b=1}^{B} \sum_{h'=1}^{H'} \sum_{w'=1}^{W'} \underline{\mathcal{I}}_{b,c_g,h,w} \Delta\mathcal{Y}_{b,c'_g,h',w'},$$

where:

$$h = h' \times \text{Stride} + k \times \text{Dilation},$$
$$w = w' \times \text{Stride} + l \times \text{Dilation},$$
$$N_{out} = \left\lfloor \frac{\text{Groups}}{C'} \right\rfloor,$$
$$N_{in} = \left\lfloor \frac{\text{Groups}}{C} \right\rfloor,$$
$$c'_g = g \times N_{out} + c',$$
$$c_g = g \times N_{in} + c,$$
$$c' \in [1, N_{out}],$$
$$c \in [1, N_{in}],$$
$$g \in [1, \text{Groups}],$$
$$k \text{ and } l \in [1, D],$$
$$\underline{\mathcal{I}} \text{ is the padded input.}$$

(17)

In our case, before being saved in memory, $\mathcal{I}$ is decomposed and compressed through truncated HOSVD as presented in (8):

$$\tilde{\mathcal{I}}_{b,c_g,h,w} = \sum_{k_1=1}^{K_1} \sum_{k_2=1}^{K_2} \sum_{k_3=1}^{K_3} \sum_{k_4=1}^{K_4} \hat{\mathcal{S}}_{k_1,k_2,k_3,k_4} \times U^{(1)}_{(K_1)_{b,k_1}} \times U^{(2)}_{(K_2)_{c_g,k_2}} \times U^{(3)}_{(K_3)_{h,k_3}} \times U^{(4)}_{(K_4)_{w,k_4}}. \quad (18)$$

Therefore, in the backward pass, its padded version can be restored using the following formula:

$$\underline{\tilde{\mathcal{I}}}_{b,c_g,h,w} = \sum_{k_1=1}^{K_1} \sum_{k_2=1}^{K_2} \sum_{k_3=1}^{K_3} \sum_{k_4=1}^{K_4} \hat{\mathcal{S}}_{k_1,k_2,k_3,k_4} \times U^{(1)}_{(K_1)_{b,k_1}} \times U^{(2)}_{(K_2)_{c_g,k_2}} \times \underline{U}^{(3)}_{(K_3)_{h,k_3}} \times \underline{U}^{(4)}_{(K_4)_{w,k_4}}, \quad (19)$$

where, $\underline{U}^{(3)}_{(K_3)}$ and $\underline{U}^{(4)}_{(K_4)}$ are $U^{(3)}_{(K_3)}$ and $U^{(4)}_{(K_4)}$ with vertical padding (top and bottom edges only), respectively. Then, we substitute (19) into (17) which gives us, after reordering and grouping:

$$\mathcal{Z}^{(1)}_{k_1,c'_g,h',w'} = \sum_{b=1}^{B} \Delta\mathcal{Y}_{b,c'_g,h',w'} U^{(1)}_{(K_1)_{b,k_1}}, \quad (20)$$

$$\mathcal{Z}^{(2)}_{k_1,k_2,h,k_4} = \sum_{k_3=1}^{K_3} \hat{\mathcal{S}}_{k_1,k_2,k_3,k_4} \underline{U}^{(3)}_{(K_3)_{h,k_3}}, \quad (21)$$

$$\mathcal{Z}^{(3)}_{k_1,k_2,h,w} = \sum_{k_4=1}^{K_4} Z^{(2)}_{k_1,k_2,h,k_4} \underline{U}^{(4)}_{(K_4)_{w,k_4}}, \quad (22)$$

$$\mathcal{Z}^{(4)}_{c'_g,k_2,k,l} = \sum_{h'=1}^{H'} \sum_{w'=1}^{W'} \sum_{k_1=1}^{K_1} Z^{(3)}_{k_1,k_2,h,w} Z^{(1)}_{k_1,c'_g,h',w'}, \quad (23)$$

$$\Delta\mathcal{W}_{c'_g,c,k,l} = \sum_{k_2=1}^{K_2} Z^{(4)}_{c'_g,k_2,k,l} U^{(2)}_{(K_2)_{c_g,k_2}}. \quad (24)$$

Computing (20), (21), (22) and (24) corresponds to performing $1 \times 1$ convolutions while (23) is a $H' \times W'$ convolution, resulting in (11).

### A.4 Details of Overhead, Computational Speedup and Space Complexity

Following the notations introduced in the main paper, we propose in this section to analitically study the overhead introduced in the forward pass when performing HOSVD compression, the speedup of efficient weight derivative computation and the required space to store the compressed activations.

**Overhead.** During the forward pass, the overhead of activation compression is the computational complexity of performing the tensor decomposition, which can be calculated as follows:

The computational complexity of SVD for a matrix of size $m \times n$ with $m \geq n$ is $\Theta_{\text{FLOPS}}(m^2 n)$. HOSVD essentially performs SVD on each mode of the tensor. During the forward pass, for each convolution layer, given an activation map of size $B \times C \times H \times W$, HOSVD involves performing SVD on four matrices of sizes $B \times CHW$, $C \times BHW$, $H \times BCW$, and $W \times BCH$. Therefore, the computational complexity for decomposition in the forward pass is:

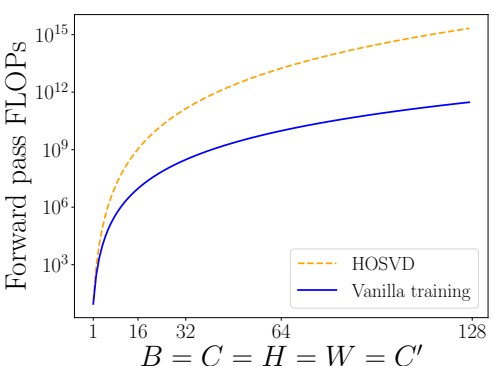

Figure 6: Predicted evolution of FLOPs for the forward pass of both vanilla training and HOSVD.

$$\text{FLOPs}_{\text{overhead}} = \Theta_{\text{FLOPS}}\Big( \max(B, CHW)^2 \times \min(B, CHW) + \max(C, BHW)^2 \times \min(C, BHW) +$$
$$\max(H, BCW)^2 \times \min(H, BCW) + \max(W, BCH)^2 \times \min(W, BCH)\Big). \tag{25}$$

Additionally, the amount of FLOPs that vanilla training requires to perform the forward pass is:

$$\text{FLOPs}_{\text{Vanilla}} = D^2 CC' BHW. \tag{26}$$

From this, the total FLOPs that HOSVD requires is:

$$\text{FLOPs}_{\text{HOSVD}} = \Theta_{\text{FLOPS}}\Big( \max(B, CHW)^2 \times \min(B, CHW) + \max(C, BHW)^2 \times \min(C, BHW) +$$
$$\max(H, BCW)^2 \times \min(H, BCW) + \max(W, BCH)^2 \times \min(W, BCH)\Big) + D^2 CC' BHW. \tag{27}$$

Based on these calculations, we represent in Fig. 6 the evolution of FLOPs in the forward pass for both vanilla training and HOSVD, showing that our method results in an increased latency in the forward pass.

**Speedup.** The key difference between HOSVD and Vanilla BP lies in computing $\Delta \mathcal{W}$.

For Vanilla BP, the cost of computing $\Delta \mathcal{W}$ is $\Theta_{\text{FLOPS}}(D^2 CC' BH'W')$.

Whereas our method involves:

Computing $Z^{(1)} : \Theta_{\text{FLOPS}}(K_1 C' H' W' B)$,

Computing $Z^{(2)} : \Theta_{\text{FLOPS}}(K_1 K_2 H K_4 K_3)$,

Computing $Z^{(3)} : \Theta_{\text{FLOPS}}(K_1 K_2 H K_4 K_3 + K_1 K_2 HW K_4)$,

Computing $Z^{(4)} : \Theta_{\text{FLOPS}}(K_1 C' H' W' B + K_1 K_2 H K_4 K_3 + K_1 K_2 HW K_4 + C' K_2 D^2 H' W' K_1)$.

Therefore, the complexity for computing $\Delta \mathcal{W}$ with our efficient reconstruction is:

$$\Theta_{\text{FLOPS}}(K_1 C' H' W' B + K_1 K_2 H K_4 K_3 + K_1 K_2 HW K_4 + C' K_2 D^2 H' W' K_1 + C' C D^2 K_2). \tag{28}$$

We thus deduce the speedup ratio $R_S$ for the backward pass, between vanilla training and HOSVD (results represented in Fig. 2b):

$$R_S = \frac{D^2 CC' BH'W'}{K_1 C' H' W' B + K_1 K_2 H K_4 K_3 + K_1 K_2 HW K_4 + C' K_2 D^2 H' W' K_1 + C' C D^2 K_2}. \tag{29}$$

**Space Complexity.** For vanilla training, storing $\mathcal{A}_i$ correspond to the storage of $\Theta_{\text{space}}(BCHW)$ elements.

For HOSVD, instead of storing the entire tensor, we store its truncated principal components: $\tilde{\mathcal{S}}$, $U_{(K_1)}^{(1)}$, $U_{(K_2)}^{(2)}$, $U_{(K_3)}^{(3)}$, and $U_{(K_4)}^{(4)}$, which corresponds to a total of $\Theta_{\text{space}}(K_1 K_2 K_3 K_4 + BK_1 + CK_2 + HK_3 + WK_4)$ elements.

Thus, we obtain the storage complexity ratio $R_C$ as follows (results represented in Fig. 2a):

$$R_C = \frac{BCHW}{K_1 K_2 K_3 K_4 + BK_1 + CK_2 + HK_3 + WK_4}. \tag{30}$$

# B    Additional Experimental Details

## B.1    Variance of Different Runs

We conducted classification experiments using MCUNet with setup A, fine-tuning on CIFAR-10, and segmentation experiments using the DeepLabV3 checkpoint provided by [53], fine-tuning on the augmented Pascal-VOC12 dataset. Both experiments employed three different random seeds (233, 234, and 235). The results, presented in Table 3 and Table 4, indicate no significant variation in performance across the different random seeds. Therefore, for subsequent experiments, we used a single random seed, 233.

Table 3: Classification with MCUNet on different random seeds.

| #Layers | Vanilla training | Gradient Filter R2 | Gradient Filter R4 | Gradient Filter R7 | SVD ($\varepsilon = 0.8$) | HOSVD ($\varepsilon = 0.8$) |
|---|---|---|---|---|---|---|
| 2 | 71.82±0.36 | 70.75±0.19 | 87.68±0.01 | 70.42±0.18 | 66.84±0.18 | 65.46±0.09 |
| 4 | 83.15±0.19 | 82.19±0.11 | 89.54±0.03 | 80.98±0.06 | 80.79±0.04 | 77.22±0.03 |

Table 4: Segmentation with DeepLabV3 on different random seeds.

| Method | #Layers | mIoU | mAcc |
|---|---|---|---|
| HOSVD ($\varepsilon = 0.8$) | 5 | 38.62±0.01 | 50.41±0.06 |
| | 10 | 50.33±0.05 | 63.39±0.05 |
| SVD ($\varepsilon = 0.8$) | 5 | 39.71±0.00 | 51.39±0.07 |
| | 10 | 52.47±0.04 | 66.09±0.01 |

## B.2    Detailed Experimental Setup

**ImageNet Classification.** In this setup, we use a similar finetuning policy to [53]. Specifically, we finetune the checkpoints for 90 epochs with L2 gradient clipping with a threshold of 2.0. We use SGD with a weight decay of $1 \times 10^{-4}$ and a momentum of 0.9. The data is randomly resized, randomly flipped, normalized, and divided into batches of 64 elements. We use cross-entropy as the loss function. The difference from their setup lies in the initial learning rate value after the warm-up epochs: in our case, the learning rate increases linearly over 4 warm-up epochs up to 0.005 (while in [53] it is 0.05). After this, similarly to their approach, the learning rate decays according to the cosine annealing method in the following epochs.

**Other dataset finetuning.** We use the same set of hyperparameters as described in [53] for both setup A and setup B. For training, we use cross-entropy loss with the SGD optimizer. The learning rate starts at 0.05 and decays according to the cosine annealing method. Momentum is set to 0, and weight decay is set to $1 \times 10^{-4}$. We apply L2 gradient clipping with a threshold of 2.0.

For setup B, batch normalization layers are fused with convolutional layers before training, a common technique for accelerating inference. We set the batch size to 128 and normalized the data before feeding it to the model.

**Semantic Segmentation.** Following the policy outlined in [53], we utilized the pretrained and calibrated checkpoints provided by the authors for our experiments. The models were pretrained on Cityscapes using MMSegmentation, then we finetuned them on the augmented Pascal-VOC12 dataset. The optimizer's learning rate starts at 0.01 and decays according to the cosine annealing schedule during training. Additionally, we set the weight decay to $5 \times 10^{-4}$ and momentum to 0.9. The batch size is set to 8. For data augmentation, we randomly crop, flip, apply photometric distortions, and normalize the images. Cross-entropy is used as the loss function. Models are finetuned for 20,000 steps.

## B.3  Additional Explained Variance Evolution Results

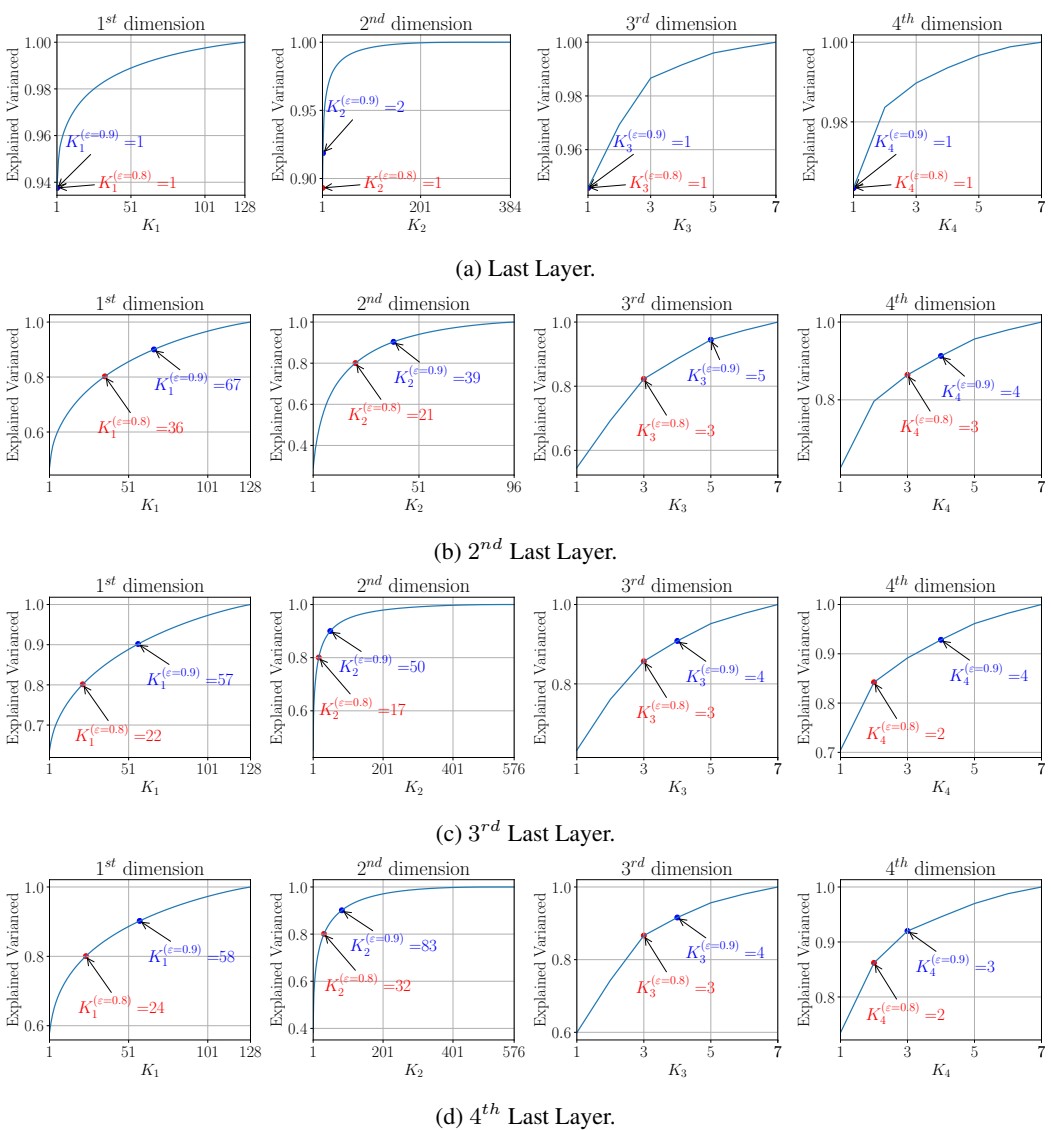

Figure 7: $K_j$ and variance for each of the last four layers of an MCUNet when fine-tuning them using HOSVD on CIFAR-10 following setup A.

As anticipated in Sec. 4.2, Fig. 7 presents the explained variance as a function of $K_j$ at the $j^{th}$ dimension of the four fine-tuned layers of MCUNet. We observe that in the first two dimensions, which are also the largest, more than $80\%$ of the tensor's energy is concentrated within the first $30\%$ of the principal components.

With the same experiment, Fig. 8 shows that $K_j$ tends to gradually increase after each training epoch, then peaks and begins to decline. Intuitively, at the beginning, the activation map energy is mainly concentrated in the few first components. As training progresses, this energy starts to spread out to other components, leading to an increase in the value of $K_j$. Over time, the model learns how to efficiently condense the energy, leading to a gradual decrease in $K_j$. This instability in $K_j$ throughout the training process is why we had to log memory as described in Section 4.1.

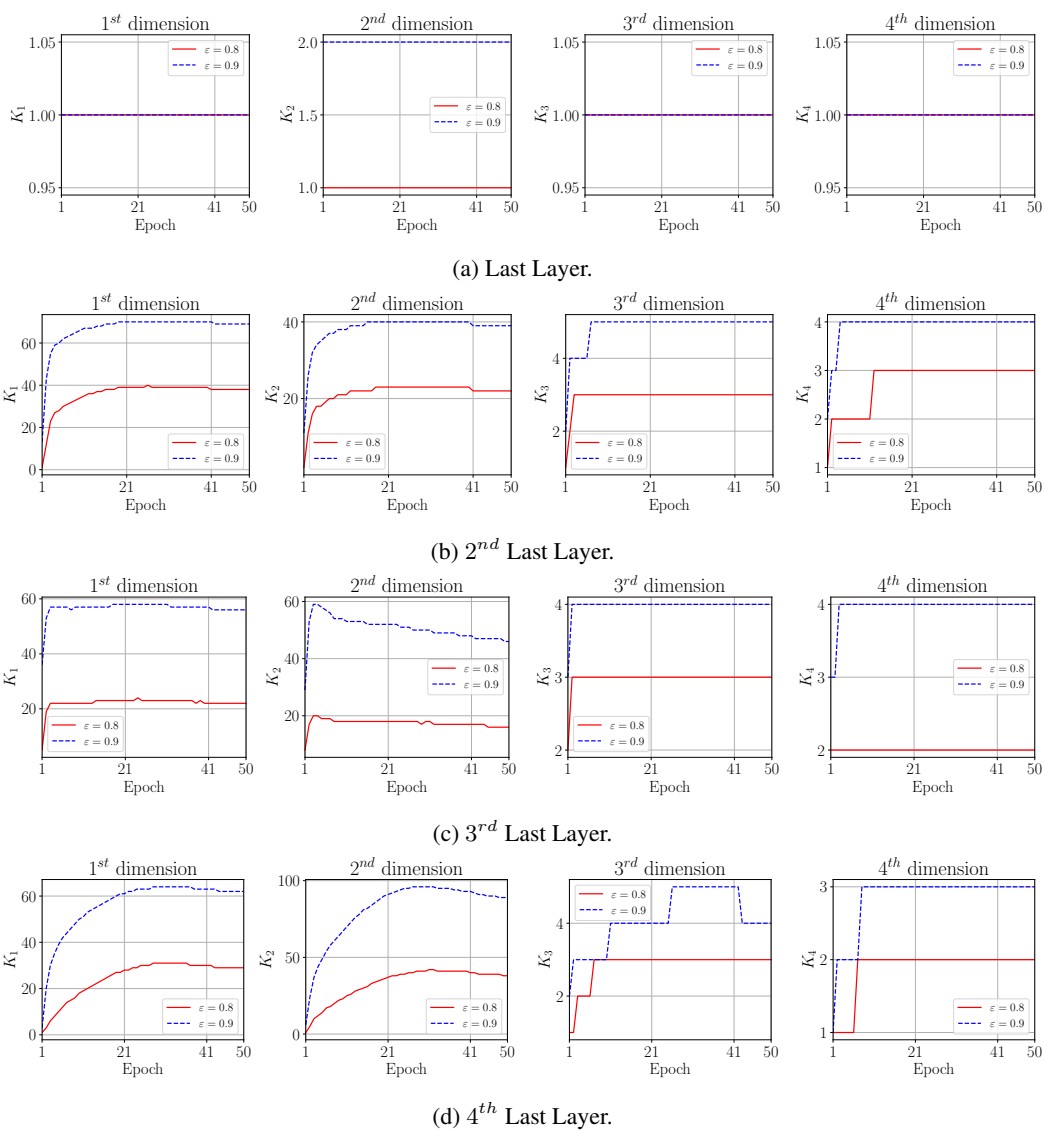

(a) Last Layer.

(b) $2^{nd}$ Last Layer.

(c) $3^{rd}$ Last Layer.

(d) $4^{th}$ Last Layer.

Figure 8: Behavior of $K_j$ during training for each of the last four layers of an MCUNet when fine-tuning them using HOSVD on CIFAR-10 dataset following setup A with two different values of $\varepsilon$.

## B.4 Processing Time Results

This section presents latency comparisons between vanilla training and HOSVD. We conducted the experiments using MCUNet on a single batch of CIFAR-10 dataset with setup A and for one epoch only.

Fig. 9 compares the execution time of the forward, backward, and total training processes between vanilla training and HOSVD. It can be observed that the backward processing time of HOSVD is tens of times lower than vanilla training, but the forward time is up to a thousand times higher. As a result, when considering the total training time (the sum of forward and backward), HOSVD is on average 4.29 times slower than vanilla training. This outcome is entirely expected, as the decomposition process introduces an overhead, as predicted and described in Fig. 6 and Appendix A.4.

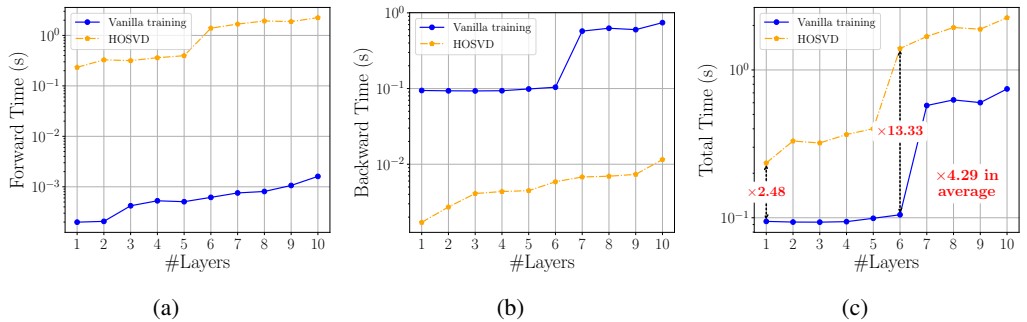

Figure 9: **(a)**, **(b)**, and **(c)** represent the time in seconds taken by the algorithm to respectively perform Forward, Backward, and the total training process, when fine-tuning an MCUNet on one batch of CIFAR-10 with Setup A across 1 to 10 layers.

Table 5: More results with setup A.

| Model | Method | #Layers | CUB200 Acc ↑ | CUB200 Peak Mem (MB)↓ | CUB200 Mean Mem (MB)↓ | Flowers102 Acc ↑ | Flowers102 Peak Mem (MB)↓ | Flowers102 Mean Mem (MB)↓ | Pets Acc ↑ | Pets Peak Mem (MB)↓ | Pets Mean Mem (MB)↓ | CIFAR-10 Acc ↑ | CIFAR-10 Peak Mem (MB)↓ | CIFAR-10 Mean Mem (MB)↓ | CIFAR-100 Acc ↑ | CIFAR-100 Peak Mem (MB)↓ | CIFAR-100 Mean Mem (MB)↓ |
|---|---|---|---|---|---|---|---|---|---|---|---|---|---|---|---|---|---|
| MobileNetV2 | Vanilla training | All | 52.9 | 3303.67 | 3303.67 ± 0.00 | 80.7 | 3303.67 | 3303.67 ± 0.00 | 89.8 | 3303.67 | 3303.67 ± 0.00 | 95.3 | 3303.67 | 3303.67 ± 0.00 | 77.9 | 3303.67 | 3303.67 ± 0.00 |
| | | 2 | 48.4 | 30.63 | 30.63 ± 0.00 | 80.3 | 30.63 | 30.63 ± 0.00 | 88.4 | 30.63 | 30.63 ± 0.00 | 88.1 | 30.63 | 30.63 ± 0.00 | 64.7 | 30.63 | 30.63 ± 0.00 |
| | | 4 | 53.1 | 57.42 | 57.42 ± 0.00 | 83.0 | 57.42 | 57.42 ± 0.00 | 89.5 | 57.42 | 57.42 ± 0.00 | 89.5 | 57.42 | 57.42 ± 0.00 | 67.6 | 57.42 | 57.42 ± 0.00 |
| | Gradient Filter R2 | 2 | 46.7 | 10.00 | 10.00 ± 0.00 | 80.9 | 10.00 | 10.00 ± 0.00 | 88.3 | 10.00 | 10.00 ± 0.00 | 87.9 | 10.00 | 10.00 ± 0.00 | 64.7 | 10.00 | 10.00 ± 0.00 |
| | | 4 | 50.2 | 18.75 | 18.75 ± 0.00 | 82.7 | 18.75 | 18.75 ± 0.00 | 89.2 | 18.75 | 18.75 ± 0.00 | 89.3 | 18.75 | 18.75 ± 0.00 | 67.2 | 18.75 | 18.75 ± 0.00 |
| | Gradient Filter R7 | 2 | 46.6 | 0.63 | 0.63 ± 0.00 | 81.0 | 0.63 | 0.63 ± 0.00 | 88.7 | 0.63 | 0.63 ± 0.00 | 86.1 | 0.63 | 0.63 ± 0.00 | 63.6 | 0.63 | 0.63 ± 0.00 |
| | | 4 | 46.4 | 1.17 | 1.17 ± 0.00 | 81.4 | 1.17 | 1.17 ± 0.00 | 88.9 | 1.17 | 1.17 ± 0.00 | 86.3 | 1.17 | 1.17 ± 0.00 | 64.6 | 1.17 | 1.17 ± 0.00 |
| | HOSVD (ε = 0.8) | 2 | 44.2 | 0.14 | 0.10 ± 0.01 | 79.1 | 0.25 | 0.07 ± 0.03 | 88.3 | 0.16 | 0.15 ± 0.01 | 85.6 | 0.30 | 0.22 ± 0.03 | 61.3 | 0.16 | 0.15 ± 0.01 |
| | | 4 | 48.9 | 0.55 | 0.54 ± 0.00 | 81.5 | 0.66 | 0.49 ± 0.03 | 89.2 | 0.73 | 0.71 ± 0.01 | 88.5 | 0.68 | 0.67 ± 0.01 | 66.1 | 0.72 | 0.68 ± 0.03 |
| | SVD (ε = 0.8) | 2 | 16.2 | 4.15 | 2.47 ± 0.57 | 23.2 | 3.85 | 2.73 ± 0.64 | 88.1 | 3.38 | 3.17 ± 0.15 | 86.4 | 3.96 | 3.08 ± 0.42 | 62.7 | 3.30 | 2.84 ± 0.30 |
| | | 4 | 16.7 | 11.34 | 9.70 ± 0.53 | 24.8 | 11.63 | 10.46 ± 0.63 | 89.1 | 12.75 | 12.50 ± 0.73 | 89.6 | 13.65 | 12.94 ± 0.76 | 67.0 | 14.10 | 13.46 ± 0.89 |
| MCUNet | Vanilla training | All | 30.8 | 1265.96 | 1265.96 ± 0.00 | 45.6 | 1265.96 | 1265.96 ± 0.00 | 75.2 | 1265.96 | 1265.96 ± 0.00 | 91.2 | 1265.96 | 1265.96 ± 0.00 | 65.6 | 1265.96 | 1265.96 ± 0.00 |
| | | 2 | 9.6 | 27.56 | 27.56 ± 0.00 | 39.3 | 27.56 | 27.56 ± 0.00 | 40.6 | 27.56 | 27.56 ± 0.00 | 72.3 | 27.56 | 27.56 ± 0.00 | 45.4 | 27.56 | 27.56 ± 0.00 |
| | | 4 | 12.4 | 39.05 | 39.05 ± 0.00 | 44.1 | 39.05 | 39.05 ± 0.00 | 49.2 | 39.05 | 39.05 ± 0.00 | 83.6 | 39.05 | 39.05 ± 0.00 | 55.3 | 39.05 | 39.05 ± 0.00 |
| | Gradient Filter R2 | 2 | 10.0 | 9.00 | 9.00 ± 0.00 | 39.1 | 9.00 | 9.00 ± 0.00 | 40.5 | 9.00 | 9.00 ± 0.00 | 71.2 | 9.00 | 9.00 ± 0.00 | 45.1 | 9.00 | 9.00 ± 0.00 |
| | | 4 | 12.3 | 12.75 | 12.75 ± 0.00 | 42.6 | 12.75 | 12.75 ± 0.00 | 49.1 | 12.75 | 12.75 ± 0.00 | 82.5 | 12.75 | 12.75 ± 0.00 | 54.2 | 12.75 | 12.75 ± 0.00 |
| | Gradient Filter R7 | 2 | 9.6 | 0.56 | 0.56 ± 0.00 | 38.3 | 0.56 | 0.56 ± 0.00 | 40.5 | 0.56 | 0.56 ± 0.00 | 70.9 | 0.56 | 0.56 ± 0.00 | 45.0 | 0.56 | 0.56 ± 0.00 |
| | | 4 | 10.3 | 0.80 | 0.80 ± 0.00 | 41.6 | 0.80 | 0.80 ± 0.00 | 44.2 | 0.80 | 0.80 ± 0.00 | 78.4 | 0.80 | 0.80 ± 0.00 | 51.0 | 0.80 | 0.80 ± 0.00 |
| | HOSVD (ε = 0.8) | 2 | 9.1 | 0.13 | 0.12 ± 0.01 | 35.6 | 0.12 | 0.12 ± 0.00 | 18.6 | 0.13 | 0.13 ± 0.00 | 65.5 | 0.07 | 0.05 ± 0.01 | 40.8 | 0.05 | 2.19 ± 0.00 |
| | | 4 | 9.6 | 7.32 | 6.06 ± 1.63 | 33.9 | 4.53 | 4.13 ± 0.21 | 46.6 | 7.56 | 6.77 ± 0.98 | 80.8 | 7.24 | 6.19 ± 1.21 | 53.7 | 6.36 | 5.55 ± 1.08 |
| | SVD (ε = 0.8) | 2 | 9.1 | 5.41 | 5.09 ± 0.33 | 36.0 | 4.77 | 4.64 ± 0.11 | 39.5 | 5.77 | 5.49 ± 0.28 | 66.9 | 2.11 | 1.65 ± 0.51 | 41.8 | 1.51 | 1.30 ± 0.28 |
| ResNet18 | Vanilla training | All | 56.5 | 1065.75 | 1065.75 ± 0.00 | 82.4 | 1065.75 | 1065.75 ± 0.00 | 88.4 | 1065.75 | 1065.75 ± 0.00 | 95.4 | 1065.75 | 1065.75 ± 0.00 | 78.6 | 1065.75 | 1065.75 ± 0.00 |
| | | 2 | 55.0 | 24.50 | 24.50 ± 0.00 | 83.8 | 24.50 | 24.50 ± 0.00 | 88.9 | 24.50 | 24.50 ± 0.00 | 91.1 | 24.50 | 24.50 ± 0.00 | 70.5 | 24.50 | 24.50 ± 0.00 |
| | | 4 | 54.0 | 61.25 | 61.25 ± 0.00 | 84.5 | 61.25 | 61.25 ± 0.00 | 88.9 | 61.25 | 61.25 ± 0.00 | 92.5 | 61.25 | 61.25 ± 0.00 | 73.3 | 61.25 | 61.25 ± 0.00 |
| | Gradient Filter R2 | 2 | 55.1 | 8.00 | 8.00 ± 0.00 | 83.7 | 8.00 | 8.00 ± 0.00 | 88.7 | 8.00 | 8.00 ± 0.00 | 90.4 | 8.00 | 8.00 ± 0.00 | 69.8 | 8.00 | 8.00 ± 0.00 |
| | | 4 | 50.2 | 14.00 | 14.00 ± 0.00 | 83.5 | 14.00 | 14.00 ± 0.00 | 88.6 | 14.00 | 14.00 ± 0.00 | 91.5 | 14.00 | 14.00 ± 0.00 | 71.6 | 14.00 | 14.00 ± 0.00 |
| | Gradient Filter R7 | 2 | 52.5 | 0.50 | 0.50 ± 0.00 | 82.1 | 0.50 | 0.50 ± 0.00 | 88.4 | 0.50 | 0.50 ± 0.00 | 88.9 | 0.50 | 0.50 ± 0.00 | 68.7 | 0.50 | 0.50 ± 0.00 |
| | | 4 | 44.8 | 0.88 | 0.88 ± 0.00 | 82.3 | 0.88 | 0.88 ± 0.00 | 86.1 | 0.88 | 0.88 ± 0.00 | 90.1 | 0.88 | 0.88 ± 0.00 | 70.1 | 0.88 | 0.88 ± 0.00 |
| | HOSVD (ε = 0.8) | 2 | 54.2 | 1.64 | 1.12 ± 0.08 | 82.4 | 1.44 | 0.84 ± 0.13 | 89.1 | 1.93 | 1.61 ± 0.07 | 90.8 | 1.50 | 1.43 ± 0.06 | 70.2 | 1.16 | 1.12 ± 0.02 |
| | | 4 | 53.2 | 3.21 | 2.38 ± 0.16 | 83.8 | 3.43 | 2.52 ± 0.22 | 88.4 | 3.84 | 3.43 ± 0.13 | 92.2 | 1.93 | 1.83 ± 0.07 | 71.8 | 1.64 | 1.47 ± 0.13 |
| | SVD (ε = 0.8) | 2 | 54.2 | 12.95 | 10.00 ± 0.62 | 82.4 | 12.69 | 9.45 ± 1.03 | 88.7 | 13.58 | 12.42 ± 0.33 | 91.0 | 12.80 | 12.69 ± 0.13 | 70.5 | 11.55 | 11.36 ± 0.17 |
| | | 4 | 53.7 | 30.91 | 26.82 ± 0.91 | 83.9 | 32.19 | 27.15 ± 1.52 | 89.1 | 32.83 | 31.08 ± 0.82 | 92.4 | 25.47 | 25.16 ± 0.16 | 72.6 | 24.63 | 24.16 ± 0.38 |
| ResNet34 | Vanilla training | All | 60.6 | 1678.25 | 1678.25 ± 0.00 | 76.9 | 1678.25 | 1678.25 ± 0.00 | 90.3 | 1678.25 | 1678.25 ± 0.00 | 96.6 | 1678.25 | 1678.25 ± 0.00 | 82.1 | 1678.25 | 1678.25 ± 0.00 |
| | | 2 | 57.8 | 24.50 | 24.50 ± 0.00 | 83.5 | 24.50 | 24.50 ± 0.00 | 90.8 | 24.50 | 24.50 ± 0.00 | 91.0 | 24.50 | 24.50 ± 0.00 | 70.4 | 24.50 | 24.50 ± 0.00 |
| | | 4 | 60.7 | 49.00 | 49.00 ± 0.00 | 83.1 | 49.00 | 49.00 ± 0.00 | 90.5 | 49.00 | 49.00 ± 0.00 | 92.3 | 49.00 | 49.00 ± 0.00 | 72.8 | 49.00 | 49.00 ± 0.00 |
| | Gradient Filter R2 | 2 | 57.5 | 8.00 | 8.00 ± 0.00 | 81.1 | 8.00 | 8.00 ± 0.00 | 90.9 | 8.00 | 8.00 ± 0.00 | 90.5 | 8.00 | 8.00 ± 0.00 | 70.0 | 8.00 | 8.00 ± 0.00 |
| | | 4 | 58.1 | 16.00 | 16.00 ± 0.00 | 82.1 | 16.00 | 16.00 ± 0.00 | 90.2 | 16.00 | 16.00 ± 0.00 | 91.6 | 16.00 | 16.00 ± 0.00 | 70.6 | 16.00 | 16.00 ± 0.00 |
| | Gradient Filter R7 | 2 | 55.6 | 0.50 | 0.50 ± 0.00 | 81.9 | 0.50 | 0.50 ± 0.00 | 90.9 | 0.50 | 0.50 ± 0.00 | 89.8 | 0.50 | 0.50 ± 0.00 | 69.6 | 0.50 | 0.50 ± 0.00 |
| | | 4 | 53.3 | 1.00 | 1.00 ± 0.00 | 81.1 | 1.00 | 1.00 ± 0.00 | 90.7 | 1.00 | 1.00 ± 0.00 | 90.7 | 1.00 | 1.00 ± 0.00 | 69.2 | 1.00 | 1.00 ± 0.00 |
| | HOSVD (ε = 0.8) | 2 | 56.1 | 0.60 | 0.38 ± 0.03 | 80.0 | 0.62 | 0.25 ± 0.08 | 90.6 | 0.70 | 0.56 ± 0.04 | 90.7 | 0.57 | 0.54 ± 0.02 | 69.9 | 0.44 | 0.41 ± 0.02 |
| | | 4 | 58.4 | 1.40 | 0.80 ± 0.12 | 81.0 | 1.27 | 0.61 ± 0.18 | 90.5 | 1.78 | 1.34 ± 0.17 | 91.6 | 1.26 | 1.16 ± 0.07 | 70.6 | 1.04 | 0.93 ± 0.09 |
| | SVD (ε = 0.8) | 2 | 56.7 | 9.99 | 6.90 ± 0.60 | 80.7 | 9.75 | 5.99 ± 1.04 | 91.1 | 10.79 | 9.50 ± 0.40 | 91.0 | 10.46 | 10.28 ± 0.11 | 70.2 | 9.41 | 9.12 ± 0.24 |
| | | 4 | 58.9 | 19.24 | 13.99 ± 1.41 | 82.7 | 20.37 | 12.82 ± 2.36 | 90.5 | 21.63 | 18.79 ± 1.33 | 92.1 | 20.84 | 20.06 ± 0.40 | 71.4 | 19.78 | 18.71 ± 0.78 |
| SwinT [26] | Vanilla training | All | 79.0 | 3748.88 | 3748.88 ± 0.00 | 89.6 | 3748.88 | 3748.88 ± 0.00 | 94.2 | 3748.88 | 3748.88 ± 0.00 | 98.0 | 3748.88 | 3748.88 ± 0.00 | 87.2 | 3748.88 | 3748.88 ± 0.00 |
| | | 2 | 61.0 | 73.88 | 73.88 ± 0.00 | 81.7 | 73.88 | 73.88 ± 0.00 | 95.4 | 73.88 | 73.88 ± 0.00 | 92.1 | 73.88 | 73.88 ± 0.00 | 74.9 | 73.88 | 73.88 ± 0.00 |
| | | 4 | 72.1 | 92.25 | 92.25 ± 0.00 | 86.9 | 92.25 | 92.25 ± 0.00 | 93.8 | 92.25 | 92.25 ± 0.00 | 94.2 | 92.25 | 92.25 ± 0.00 | 78.2 | 92.25 | 92.25 ± 0.00 |
| | HOSVD (ε = 0.8) | 2 | 55.1 | 1.91 | 1.85 ± 0.02 | 74.9 | 2.42 | 2.36 ± 0.02 | 93.1 | 4.01 | 3.93 ± 0.04 | 91.6 | 5.52 | 5.41 ± 0.11 | 73.5 | 6.39 | 6.19 ± 0.23 |
| | | 4 | 67.4 | 1.55 | 0.65 ± 0.17 | 82.5 | 2.58 | 1.21 ± 0.48 | 93.4 | 3.01 | 2.48 ± 0.11 | 93.8 | 6.00 | 5.65 ± 0.41 | 77.2 | 5.84 | 5.35 ± 0.73 |
| | SVD (ε = 0.8) | 2 | 55.3 | 35.85 | 35.65 ± 0.11 | 75.2 | 36.78 | 36.57 ± 0.09 | 93.1 | 40.75 | 40.36 ± 0.16 | 91.6 | 46.08 | 45.90 ± 0.19 | 73.5 | 47.37 | 47.05 ± 0.38 |
| | | 4 | 68.5 | 40.23 | 27.01 ± 2.85 | 82.6 | 43.78 | 35.71 ± 3.34 | 93.4 | 45.58 | 44.02 ± 0.39 | 93.9 | 56.62 | 56.09 ± 0.70 | 77.3 | 57.38 | 56.31 ± 1.84 |

## B.5 Additional Classification Results

In addition to the main experiments, we also performed many classification experiments with setup A on various datasets, the results are shown in Table 5. Especially, when fine-tuning SwinT on CUB200, Pets, and CIFAR-100, applying HOSVD to the last four layers (maintaining the rest of the architecture frozen) consumes less memory than fully fine-tuning the last two layers. This happens because the variance of the activation maps is concentrated in few components: for the same $\varepsilon$, $K$ will be smaller.

## C Limitation

While HOSVD is effective in compressing the forward signal in DNNs for gradient estimation during backpropagation, it is just one among possible choices when dealing with tensor decomposition. Indeed, more variants of tensor decomposition methods such as GKPD or STP could have been used and evaluated and could have led to larger memory and computational gains. Our work opens the door to further explorations in such direction.

