# OpenReview forum: "Activation Map Compression through Tensor Decomposition for Deep Learning"
_NeurIPS.cc/2024/Conference — NeurIPS 2024 poster_

### Official Review · Reviewer_WgAz · 2024-07-05

**Soundness:** 2
**Presentation:** 3
**Contribution:** 2
**Rating:** 3
**Confidence:** 5

**Summary:**

The paper addresses the challenge of on-device training for deep learning models, particularly focusing on the memory bottleneck caused by the storage of activation maps during backpropagation. The authors propose a method to compress activation maps using tensor decomposition techniques, specifically Singular Value Decomposition (SVD) and its tensor variant, High-Order Singular Value Decomposition (HOSVD).

**Strengths:**

1. Using tensor decomposition for activation map compression to address a bottleneck in on-device training.
2. The paper provides a solid theoretical foundation, including error analysis and guarantees for convergence.

**Weaknesses:**

1. Using SVD or tensor decomposition to compress activation maps incurs significant computational overhead. This approach of trading computational complexity for space complexity is questionable, especially considering that the computational capacity of embedded devices is usually also limited. Moreover, this compression may introduce additional information loss, potentially affecting the model's performance.
2. For SGD, recalculating activations instead of storing them is a viable method to save memory. Given that both methods trade computation time for storage space, why use low-rank compression which may potentially affect model performance?
3. With limited computational resources, why not search for a more appropriate model architecture using NAS?

**Questions:**

1. LOMO[1] fused gradient computation and parameter update in one step to minimize the size of gradient tensors. Compared to this approach, what are the advantages of your method?
2. Would using quantization methods instead of compression methods produce better results?


[1] Lv, Kai, et al. "Full parameter fine-tuning for large language models with limited resources." arXiv preprint arXiv:2306.09782 (2023).

---

> ### Author Rebuttal · Authors · 2024-08-06
>
> **[W1.1: Compression cost]** Please refer to answer #3 of *General answers*.
>
> **[W1.2: Introduced compression loss]** Please refer to answer #4 of *General answers*.
>
> **[W2: Comparison with activation checkpointing]**  As evidenced in Figure 1 of the PDF rebuttal file, the required memory to store the compressed activations of all the layers of an MCUNet is far below the full activation memory of the last layer of that same network. This is especially meaningfull as that specific layer is one of the least costly in terms of activation memory compared to other layers in the network. In that sense, although activation checkpointing results in an equivalent accuracy to that of vanilla BP, its memory consumption is about one to two orders of magnitude above low-rank compression applied to all the layers. This truly underlines the idea that the accuracy tradeoff of low-rank compression is viable in the case of training on edge devices with extremely limited memory.
>
> **[W3 & Q2: Comparison with NAS and quantization]** These research directions are orthogonal to what we propose. We intend to study the effect of activation compression through tensor decomposition in a vacuum. Future research will lead us to combine this method with other compression strategies such as quantization or memory-efficient networks to further validate its generalizability. Furthermore, MCUNet, a model that we use for our experiments, was designed through NAS [24].
>
> **[Q1: Comparison with LOMO]**  As exposed in eq. (2) of our paper, activations are necessary to compute weight derivatives and they must be stored in CPU memory during the forward pass, meaning that our method could still be relevant in conjunction with LOMO to further reduce CPU and GPU memory usage. LOMO combines gradient computation and weight update in a single step instead of computing all gradients then updating all weights which allows for reduced GPU memory usage. In that sense, LOMO and low-rank activation compression adress different aspects of BP acceleration and memory reduction which means that they could be combined in an unified framework for improved savings.

---

> > ### Comment · Reviewer_WgAz · 2024-08-13
> >
> > Thank you for providing a detailed rebuttal. What I want to know is, under limited memory conditions, which method has the least negative impact on model performance or even improves it? If NAS has already identified a network that meets resource constraints, why would you still opt for compressing activations, a method that can potentially degrade performance? Additionally, I didn’t see a comparison between this method and using checkpointing. Additionally, why not choose to reduce memory consumption by swapping storage instead? Thank you for your response.

---

> > > ### Author Response · Authors · 2024-08-13
> > > **Answer to Reviewer WgAz concerns**
> > >
> > > Thank you for your response. Please find the answers to your concerns below.
> > >
> > > [**What I want to know is, under limited memory conditions, which method has the least negative impact on model performance or even improves it?**]
> > > HOSVD is the method that has the least negative impact on model performance for the same memory budget (and in the very low memory budget regime under exam).
> > > Specifically, in Figure 1 in the Rebuttal file, the horizontal axis represents activation memory, and the vertical axis represents Top-1 validation accuracy. The first data point of a curve corresponds to fine-tuning the last layer, the second point corresponds to fine-tuning the last two layers, and so on until all layers are fine-tuned. The best method is the one with a Pareto curve that leans towards the **top left** corner of the graph (indicating a low memory usage and a high Top-1 validation accuracy). Among the methods compared, the one with the Pareto curve closest to this position is HOSVD.
> > >
> > > [**If NAS has already identified a network that meets resource constraints, why would you still opt for compressing activations, a method that can potentially degrade performance?**]
> > > Because under the same memory constraint, NAS requires a *significant offline computational cost* to find an optimal solution, while our method can easily address this issue directly online (on-device).
> > > We remark that the potential loss is also tunable through a hyperparameter ($\varepsilon$).
> > > Finally, under the same memory constraints, combining our method with NAS obviously can expand the search space by relaxing memory constraints hundreds of times, leading to even more optimal models. Our approach can be integrated with NAS approaches.
> > >
> > > [**Additionally, I didn’t see a comparison between this method and using checkpointing.**]
> > > In any case, the checkpointing would be worse than HOSVD.
> > > In terms of memory, the best scenario for activation checkpointing is exactly the case when fine-tuning only the last layer with vanilla BP (>$10^4$kB in Fig. 1 of the rebuttal file). That is also one of the least memory-expensive layers, so we can expect that checkpointing would consume considerably more memory than HOSVD, making it in general worse. We agree that including this curve can be an interesting comparison, and *we commit to doing it in the final version of the paper*.
> > >
> > > [**Additionally, why not choose to reduce memory consumption by swapping storage instead?**]
> > > Because we expect offloading data to other storage units will *significantly* increase latency. For this reason, we want to design a technique that can directly fit the model in memory and we compare with similar approaches not relying on external memory sources. However, we agree that this is an interesting baseline, and *we will include the time overheads of the memory transfers on real devices in the final version of the paper*.

---

### Official Review · Reviewer_dqZf · 2024-07-08

**Soundness:** 3
**Presentation:** 3
**Contribution:** 3
**Rating:** 4
**Confidence:** 4

**Summary:**

This paper proposes a method to compress activation maps in deep neural networks using tensor decomposition techniques, specifically Singular Value Decomposition (SVD) and Higher Order SVD (HOSVD). The goal is to reduce memory requirements during backpropagation, enabling on-device learning for resource-constrained environments. The authors provide theoretical analysis of their method's impact on memory usage, computational complexity, and error bounds. They demonstrate the effectiveness of their approach through experiments on various tasks, architectures, and datasets.

**Strengths:**

- Novelty: The use of tensor decomposition for compressing activation maps is a novel approach that addresses a significant bottleneck in neural network training.
- Efficiency: The method significantly reduces memory usage during backpropagation, which is crucial for deploying deep learning models on resource-constrained edge devices.
- Theoretical Support: The paper provides  theoretical background for the proposed method, including error analysis and guarantees of minimal information loss.

**Weaknesses:**

- Severe performance degradation relative to memory reduction
- Complexity of Implementation: Implementing tensor decomposition techniques like HOSVD can be complex and may require specialized knowledge, potentially limiting its adoption.
- Lack of Detailed Rank Selection: The paper does not provide sufficient explanation on how the appropriate rank for decomposition is selected, which is crucial for balancing memory reduction and accuracy.
- Inconsistent and contradictory results: SVD frequently outperforms HOSVD, contradicting the authors' hypothesis and lacking adequate explanation.
- Limited comparison with other activation map compression techniques.

**Questions:**

- How does the proposed method perform on more diverse and complex neural network architectures, such as transformers?
- What are the practical challenges and considerations when deploying this method on real-world edge devices with strict latency and power constraints?
- How is the appropriate rank for decomposition selected, and what guidelines can be provided for tuning this parameter effectively?

**Limitations:**

- Specific to Certain Architectures: The experiments focus mainly on specific neural network architectures and tasks. It is unclear how well the method generalizes to other types of models and applications.
- Dependence on Hyperparameters: The effectiveness of the compression is highly dependent on the chosen hyperparameters for the decomposition, which may require extensive tuning.
- Resource Requirements for Decomposition: While the method reduces memory usage during training, the initial decomposition itself can be computationally intensive and may require powerful hardware.
- Practical Usability Concerns: Despite significant memory reduction, the method results in considerable accuracy drops, raising concerns about its practical usability. Vanilla backpropagation, while using more memory, generally maintains higher performance and might be a more realistic approach for actual deployment.

---

> ### Author Rebuttal · Authors · 2024-08-06
>
> **[W1: Severe performance degradation relative to memory reduction]**
> We are focusing on performing direct training on edge devices with extremely limited memory, so it is worth considering trading off accuracy for memory savings at an incredible rate.
>
> **[W2: Implementation complexity]**
> We have the implementation, and we commit to publish it as open source upon acceptance of the paper.
>
> **[W3 & Q3: Lack of detailed rank selection]**
> Please refer to answer #4 of *general answers*.
>
> **[W4: SVD vs HOSVD]**
> Please refer to answer #2 of *general answers*.
>
> **[W5: Activation map compression comparison]**
> Please refer to answer #1 of *general answers*.
>
> **[Q1: Alternative architectures]**
> We provide results on SwinT architecture in table 5 in supplementary materials sec B.2.1. For every downstream task, HOSVD allows for incredible compression rates (up to 40 times less memory than vanilla backpropagation on 2 layers and 20 times less than SVD). Regarding accuracy, HOSVD is competitive with SVD and depending on the dataset, can be slightly below vanilla BP for the same number of fine-tuned layers. These results are on par with the results obtained with the CNN architectures.

---

> > ### Comment · Reviewer_dqZf · 2024-08-11
> >
> > Thank you to the authors for the clarifications. These efforts have significantly enhanced the quality of the research. However, I believe my initial evaluation remains valid. Therefore, my score remains unchanged.

---

> > > ### Comment · Area_Chair_WUjK · 2024-08-11
> > >
> > > Dear Reviewer dqZf,
> > >
> > > Thank you for responding to the author rebuttal.
> > >
> > > >  However, I believe my initial evaluation remains valid.
> > >
> > > Can you elaborate this point? From the above comment, it is not clear to me which parts of your initial evaluation are addressed/not addressed.
> > >
> > > Please clarify that and explain why you think so. That will help the authors improve their work if it's the case.
> > >
> > > Best,
> > > Your AC

---

### Official Review · Reviewer_HEKd · 2024-07-12

**Soundness:** 3
**Presentation:** 2
**Contribution:** 3
**Rating:** 7
**Confidence:** 3

**Summary:**

[Editing to reflect my score increase from 6 to 7 after the author discussion phase.]

The authors tackle the problem of memory consumption due to needing to keep realized activation tensors available between their use in the forwards pass and backwards propagation in training neural networks.  They propose the use of tensor decomposition, specifically SVD and higher-order SVD, to compress activation maps and show that the compressed activations can be used in backpropagation without first decompressing.  An error analysis is performed, and experimental results confirm the hypothesis that significant compression can be applied while maintaining significant explained variance.  To validate the method's success on real workloads, the authors apply it to image classification and semantic segmentation tasks, fine-tuning pre-trained models for both tasks, as well as training from scratch for image classification.

**Strengths:**

**Originality**
I believe this is a novel application of tensor decomposition, a well-known technique previously applied to weight compression.  Adaptively sizing the decomposed tensor to maximize retained information in the activation maps is an advancement over past work.

**Quality**
The authors have performed a reasonable set of experiments to show the technique works as intended: model quality when activations are subject to tensor decomposition for backpropagation remain competitive for image classification tasks and semantic segmentation tasks.  Further, the authors have reported the memory consumption, proving that it is reduced significantly enough to satisfy constraints of edge devices in many cases.  I greatly appreciate the time spent in the background, as well as the seemingly offhand note that any error induced by these decompositions will be limited to that layer's weight gradients and does not accumulate deeper into the network.

**Clarity**
I generally found the manuscript easy to follow: the overall organization was great, and the method itself was explained adequately.

**Significance**
The intended use case aside, I think even datacenter users may be interested in this technique, as many of today's latest networks are simply huge.  Reducing the storage costs for activations between forward and backwards passes could reduce the parallelism needed, and in turn the overhead of communications in distributed training.  I'd encourage the authors to consider this angle and apply the technique to, for example, large language models.

**Weaknesses:**

**Originality**
Only one piece of past work comes to mind that was omitted: Rhu et al., "Compressing DMA Engine: Leveraging Activation Sparsity for Training Deep Neural Networks" (HPCA 2018) compresses activation maps, using their existing sparsity, to reduce the cost of offloading in order to accelerate training.  While novel hardware was designed to keep up with compute throughput, if compression is the only goal, then this could simply be performed in software (like the submission's tensor decomposition).

**Quality**
A missing piece is the cost of compression: what impact does this have on the training speed?  This is related to the claim on line 59 that "gradient computation and parameter updating are considerably more expensive than the forward pass," which isn't obvious to me.  At a high level, each of the three calculations (forward propagation, gradient calculation, weight update calculation) are roughly equivalent in cost, and applying the weight update is very simple.  Increasing the cost of any one of the phases would seem to be as painful as any of the others.  Another claim that stood out, given the stated goal of "reducing the memory required for backpropagation" (line 72) is that the "main challenge limiting the feasibility of on-device learning lies in the computational cost of backpropagation" (line 58).  I'd also question the claim that "we can assume that the networks considered are already [weight] compressed" (line 159) - while it's extensively studied, it's not a solved problem.  It's fine to not tackle this particular problem, but stating that it's assumed to not be an issue is misleading.

Some of the results deserve some extra discussion.  Do the authors have a hypothesis about why SVD is superior to HOSVD (line 288)?  Why does MobileNetV2's Vanilla BP results get worse when including 4 layers compared to only 2 (Table 2)?

I have not seen CIFAR treated as a "downstream task" of ImageNet-1k before; it's typically just a simpler data set used, recently, as a proving ground before testing on larger data sets.  Is it common to fine-tune a network trained on ImageNet-1k on CIFAR10/100?

Given the significant memory savings of the technique, I'd have liked to see how it performs when fine-tuning all layers, not just a subset.  Is there a reason this result wasn't gathered?

**Clarity**
I've had to make an assumption about the behavior of the method: during forward propagation, I assume the *uncompressed* activations are used, and only compressed when writing to memory to be used later in the backwards pass.  (If this is not the case, then errors *will* compound.)

Figure 2 shows speedups and space gains, but without indicating what, exactly, is speeding up or seeing a reduction in memory space.  I assume, but am not sure, that it is for a single weight gradient calculation (of a size noted in the figure's legend and axes).

The wording of Figure 3's caption is confusing: I do not understand the meaning behind "Mbv2 fine-tuned on Cifar10 at initialization."

None of the networks used in the experimental results have references attached.  For example, MCUNet is prominently used in Figure 4, but I couldn't find either a description of this network or a reference to it in the text.  It looks likely that it is in reference [24], but this only has a passing citation in the related work section.

There are some typos:

- Line 223: "derives" -> "derivatives"

- Line 234: the range of explained variance uses two open brackets.

- Reference [48] is listed with the incorrect year: 2024 should be 2023.


**Significance**
There are several ways that the results would be more compelling:

- Evaluation on more (difficult) tasks, like language modeling or object detection.

- Application to more recent or larger networks.

- More closely matching the densely trained or fine-tuned baselines.  As this is initial work in this area, the results shouldn't be dismissed, but there's still a non-trivial gap between the tensor-decomposed results and what the networks are capable of learning.

**Questions:**

1. Is there a reason that results for fine-tuning all layers were omitted?
2. Why is SVD superior to HOSVD in some results (line 288)?
3. Why does MobileNetV2's Vanilla BP results get worse when including 4 layers compared to only 2 (Table 2)?
4. What impact does the compression process have on the training speed?

**Limitations:**

The authors have addressed the limitations (in the appendix).

---

> ### Author Rebuttal · Authors · 2024-08-06
>
> **[W1 - Originality: Missing reference]** In their work Rhu et al. exploit inherent activation sparsity for compression, resulting in low memory usage and accelerated training. This is a very relevant paper with respect to our research and we will cite it in our paper, thank you for pointing it out. It is worth noting that we cite in our related works section Kurtz et al. work on activation sparsity [22]. In their work they mention Rhu et al. work as a reference and build upon their findings to induce improved activation sparsity.
>
> **[W2.1 - Quality & Q4: Cost of compression]** Please refer to answer #3 of *general answers*.
>
> **[W2.2 - Quality: Backpropagation vs Forward complexity]** Thank you for your feedback.
>
> Consider convolutional layer $i$ with:
> - An activation map $\mathcal{A}_i \in \mathbb{R}^{B\times C\times H\times W}$,
> - A weight tensor $\mathcal{W}_i \in \mathbb{R}^{C'\times C\times D\times D}$,
> - Produces an output $\mathcal{A}_{i+1} \in \mathbb{R}^{B\times C'\times H'\times W'}$.
>
> During the training process of a convolutional deep learning model:
> - There is one convolution operation in each forward pass with the computational complexity is $\mathcal{O}_{\text{time}}(D^2CC'BH'W')$.
> - While, in each backward pass, there are two convolution operations, corresponding to formulas (2) and (3) in our paper and one weight update operation, having computational complexities of:
>     - $\mathcal{O}_{\text{time}}(D^2CC'BH'W')$
>     - $\mathcal{O}_{\text{time}}(D^2CC'BHW)$
>     - $\mathcal{O}_{\text{time}}(D^2CC')$
>
> $\Rightarrow$ Therefore, we can calculate the computational complexity ratio between the forward and backward passes. This ratio is always less than 1, indicating that the backward pass is more complex. This supports our claims, and we will include this information in our supplementary materials for clarity.
>
> **[W2.3 - Quality: Weight compression SOTA]** We understand how our claim might be misleading and we will change line 158-159 to: “Weight compression is an extensively explored matter for network acceleration and we do not intend to further this area of research in this work.”
>
> **[W3.1 - Quality & Q2: SVD vs HOSVD]**
> Please refer to answer #2 of *general answers*.
>
> **[W3.2 - Quality & Q3: MbV2 Vanilla BP result]** In our paper, we report results extracted from the paper Efficient On-device Training via Gradient Filtering to compare with our method. However, we have recently reproduced their experiments using the same setup and obtained different, more reasonable results. Specifically, for Vanilla BP with 2 layers, the accuracy is now 62.6%, while it is 65.8% for 4 layers.
>
> **[W3.3 - Quality: C10/100 as downstream tasks]** We used CIFAR10/100 as downstream tasks in a similar fashion to what has been introduced in [49] as they also feature these datasets in their results. As an additional note, Han et al. use CIFAR10/100 as downstream tasks in their works [2] and [25].
>
> **[W3.4 - Quality & Q1: Fine-tuning all layers]** We will provide the complete set of results and will include them in the final version of the paper. The full results are presented in Figure 1 and Table 1 in the PDF rebuttal file.
>
> **[W4.1 - Clarity: Method Clarity]** You are correct. This can be inferred from Figure 1 of our paper. However we agree that it has not been made sufficiently clear in the description of the method. In Section 3.3 "Backpropagation with Compressed Activation," we will include the following paragraph: "Figure 1 illustrates our method. At training time, forward pass proceeds as usual with the only difference that instead of storing full activation maps in memory, we propose to store their principal components, which are the products of the decomposition process. During the backward pass, the principal components are retrieved from memory and used for calculations as described in formulas (18) to (22).”.
>
> **[W4.2 - Clarity: Figure 2 clarification]** Yes, it is for a single layer. We will make the graph easier to understand by adding more detailed descriptions.
>
> **[W4.3 - Clarity: Clarity in figure 3]** In this figure, four components are essential: the fine-tuning dataset, the network considered, the layer within the network from which we extract the explained variance curves, and the epoch at which we extract the curves. The “at initialization” means that these plots are extracted at epoch 0. We will change the caption of the figure to better reflect that aspect.
>
> **[W4.4 - Clarity: Networks references]** We use the same networks as in [49]. We will add proper referencing to all the networks considered in this paper.
>
> **[W4.5 - Clarity: There are some typos]** Thank you for your remark, we will fix them in the camera-ready version.

---

> > ### Comment · Reviewer_HEKd · 2024-08-09
> > **Comments appreciated, two remaining questions**
> >
> > W1: A reference-by-reference can be acceptable in cases where the later reference supersedes the former.  In light of the different target domains (training vs. inference) and your claim that "With the exception of Eliassen et al.’s work which accelerates training runtime, most of these works focus on accelerating inference, in a similar way to traditional model compression," then I think an explicit reference and inclusion is warranted, so thank you for the update.
> >
> > W2.1: What are the units of overhead in Figure 3(c)?
> >
> > W2.2: I understand, now - your "gradient computation and parameter updating" includes all the operations in backprop.  Thank you for the clarification.
> >
> > W2.3: Sounds like a good revision, thanks.
> >
> > W3.1: I see - for a given memory budget, HOSVD is superior to SVD.  It might make this argument more clear if the SVD and HOSVD results in Table 2 resulted in either the same accuracy or the same memory consumption (or if HOSVD were superior in both metrics).  Further, if there's an explanation for something the authors find "Suprising" (line 288), then including the explanation for this phenomenon will help the reader understand the behavior.
> >
> > Figure 4 in the submission looks like it has different data than Figure 1 in the rebuttal PDF.  For example, HOSVD (red curve) at 10^2 kB peak memory has an accuracy of ~84% in Figure 4, but only ~66% in Figure 1.  Is this a different experiment, or has something else changed?  The Gradient Filter curves appear unchanged.
> >
> > W3.2: I see, thank you for the explanation and update.  Please note in tables and figures where the data presented is from another source or your own implementation.
> >
> > W3.3: I cannot see in [49] where a model trained with the ImageNet1K is applied to a CIFAR data set; I see different models trained for either data set, but perhaps this detail is omitted since it's so common?  Thank you for pointing out the transfer learning application of ImageNet1K->CIFAR10/100 in [2] and [25].
> >
> > W3.4: Thank you for these additions.
> >
> > W4.1: Thank you for the confirmation, and the proposed revision looks great.
> >
> > W4.2: Thank you for the confirmation.
> >
> > W4.3: I see, so this reflects one input sample from CIFAR10 (on a network pre-trained with ImageNet1K?).
> >
> > Embedded above are two remaining questions:
> > 1) What are the units of overhead in Figure 3(c)?
> > 2) Why does rebuttal Figure 1 differ so much from submission Figure 4?

---

> > > ### Author Response · Authors · 2024-08-11
> > > **Thank you, answer to the extra two questions**
> > >
> > > Many thanks for your prompt response. Here below please find answers for the embedded questions.
> > >
> > > [**Units of overhead in Fig. 3(c)**]
> > >
> > > The computation overhead is here measured in FLOPs. Specifically, the x-axis represents the assumed size of the tensor that needs to be decomposed, while the y-axis shows the overhead of the forward pass, which is the computational complexity of HOSVD for decomposing that tensor (as we discussed in General answer #3).
> > >
> > > [**Rebuttal Fig. 1 different from paper’s Fig. 4**]
> > >
> > > Many apologies for the confusion around this point. That's because we have updated the plot in the Rebuttal Fig. 1 (we will replace Fig. 4 in the paper with this one). Specifically:
> > > - For HOSVD/SVD, we used to record the activation memory in the first epoch, but we changed it for the peak activation memory throughout the training process for a more accurate comparison as this is a key aspect in memory-constrained environments. The change in memory occupation is possible given that the number of components is not fixed but depends on the variance $\varepsilon$, causing a shift to the right of the curves. Please note as well that we have added more points for both the curves, and that the x axis covers a larger range of values (because of the addiction of the Vanilla BP). We remark that all the other memories presented in the paper are calculated on the *peak occupation throughout the full training*.
> > > - The Gradient Filter curves remained unchanged.
> > > - The curve on Vanilla BP has been added.

---

> > > > ### Comment · Reviewer_HEKd · 2024-08-13
> > > > **Quality and clarity greatly improved**
> > > >
> > > > Thank you, authors, for your responses.  I find my concerns regarding the quality and clarity of the submission largely addressed, and significance is the main remaining weakness.  I'll therefore raise my score to 7 - Accept.
> > > >
> > > > One note: the FLOPs overhead figure may be more impactful if the baseline computation amount is also shown, or if the overhead is expressed as a percentage of the baseline computation.  Without quickly seeing how many FLOPs are required for the un-modified  forward pass, the FLOPs required for the decomposition aren't as informative as they could be.

---

### Official Review · Reviewer_f1cQ · 2024-07-23

**Soundness:** 2
**Presentation:** 2
**Contribution:** 2
**Rating:** 4
**Confidence:** 5

**Summary:**

Tensor low-rank decomposition is used to compress the backpropagation process of full connection and convolution in neural networks. The main process is to decompose input activation tensors into Tucker structures using HOSVD algorithm and truncate subtensors at each mode. Experimental result find an efficient trade-off between the desired explained variance and compression rates.

**Strengths:**

(1) Compute the approximated weight derivatives without reconstructing the activations, through the successive computation of simpler convolutions.

(2) The experiments on trade-off between the desired explained variance and compression rates are sufficient.

(3) Experimental results obtained on mainstream architectures and tasks show that it has a Pareto advantage over other comparison schemes in terms of the trade-off between accuracy and memory footprint.

**Weaknesses:**

(1) There are a lot of confusing expressions on the formula.

(2) In formula 8, How to distinguish $U^{(k_j)}$s when the size retained after truncation on two modes is the same?

(3) The symbol $]$ in formula 11 is in the wrong position, and $\frac{\partial A_{i+1}}{\partial W_i}$ should be $\frac{\partial L}{\partial A_{i+1}}$?

(4) In Sec3.4 $\frac{\partial L}{\partial A_{i+1}}=\Delta Y$, but in appendix A.2 it expressed as \Delta O.

**Questions:**

(1) In formula 12, how to use the discrete Fourier transform to get $\tilde{I} =\varepsilon I$? What does $uI$in the denominator mean?

(2) In formula 16, please give the definitions of $d, h’,w’$.

(3) The methods mentioned in the related work are not compared with the proposed methods.

**Limitations:**

If the contraction between matrix and tensor is replaced by 1*1 convolution, then the proposed method should also be compared with CP decomposition, tensor train decomposition and other related methods.

---

> ### Author Rebuttal · Authors · 2024-08-06
>
> **[W1: Confusing formulas]**
> Thanks for your feedback, we are taking into account your comments to improve the readability.
>
>
> **[W2: Distinguishing $U^{(k_j)}$s]**
> We acknowledge the notation issue, taking action to fix it. Specifically, we will change the notation for factor matrices introduced in eq (6) from $U^{(j)}$ to $U_j$. The truncated version will thus become $U_j^{(k_j)}$. This will allow for distinction of the $U^{(k_j)}$ after truncation.
>
> **[W3: Mistakes in formula 11]**
> Thank you for your remarks, they are both correct. We will update the paper accordingly.
>
> **[W4: Inconsistent notation]**
> We agree on the remark, we will correct it in the appendix to ensure consistent notations in the final version of the paper.
>
> **[Q1: Fourier transform formula 12]**
> In formula 12, $vI$ in the denominator should be replaced by $\varepsilon I$ (sorry for our typo). By construction, $\tilde{I}$ contains the $k$ principal components of $I$ such that it explains for a percentage $\varepsilon$ of the variance of $I$. By switching to the discrete frequency domain, $I$ and $\tilde{I}$ become power spectrums where $\tilde{I}$ is composed of $k$ components originating from $I$. The sum of these components amount to a fraction of the power of the original signal which by construction is $\varepsilon$, thus $\tilde{I}=\varepsilon I$.
>
> **[Q2: Definition of $d,h',w'$]**
> We are sorry, these definitions were missing in the paper. We have $d\in [1, D]$, $h'\in [1, H']$ and $w'\in [1, W']$ where $D$ is the weight kernel dimension, $H'$ and $W'$ are the output activation height and width respectively.
>
> **[Q3: Comparison with related works methods]**
> Please refer to answer #1 of *general answers*.

---

### Author Rebuttal · Authors · 2024-08-06

First of all, we would like to thank you and express our appreciation for the time and effort you have invested in reviewing our work. We are especially grateful for the highlights regarding the novelty of the proposed method (HEKd, dqZf), the theoretical groundings (HEKd, dqZf, WgAz) and the experimental results showing its efficiency (f1cQ, HEKd). Below, we have provided detailed responses to your questions and concerns, together with the PDF Rebuttal file containing additional figures and a table. We look forward to further feedback during the discussion period. Thank you once again for your valuable input.

**Note:**
- Wi: Weakness number i
- Qi: Question number i

# General answers

**#1: Related works comparison (answer for Q3 of f1cQ and W5 of dqzf)**

Regarding tensor decomposition, methods introduced in the related works section either focus on weight compression for accelerated inference and lighter networks [13, 50, 52], or on gradient compression for accelerated backpropagation [48] whereas our approach compresses activation maps for efficient and lightweight backpropagation, thus making them not directly comparable.
In a similar fashion most current Activation Map Compression strategies focus on accelerated inference [10, 12, 22]. [9] apply their method for large GNN training. We considered this setup to be too far remote from our own experimental setup and we leave comparison with this work for future research.
In the context of on-device learning, we compared our technique to the state-of-the-art method, Gradient Filtering [49]. Other methods introduced in this paragraph focus on efficient subgraph selection [23, 25, 33] which is orthogonal to our proposition and thus compatible for joint implementation. We leave this research area for future work.

**#2: Discussion on results (answer for W3.1 - Q2 of HEKd and W4 of dqZf)**

 - **Discussion**: SVD is only slightly better than HOSVD in terms of accuracy but is significantly worse in terms of memory usage. Specifically, Figure 4 in our paper shows the Pareto curves for SVD, HOSVD, and Gradient Filter. It is clear that HOSVD has a significantly better Pareto curve. To be clearer, we will update Figure 4 by Figure 1 in the PDF Rebuttal file showing the full pareto curve of HOSVD and SVD.
- **Explanation**: HOSVD essentially performs SVD on each mode of a tensor. Specifically, applying HOSVD to a tensor $\mathcal{A} \in \mathbb{R}^{B \times C \times H \times W}$ involves performing SVD on four unfolded versions of $\mathcal{A}$ with sizes $B \times CHW$, $C \times BHW$, $H \times BCW$, and $W \times BCH$, respectively. In contrast, SVD is applied only to the first mode, which is a tensor of size $B \times CHW$ (lines 180 - 181 in our paper). Thus, with SVD, the explained variance threshold is applied to retain an $\varepsilon$ fraction of the information in the first mode, **without affecting other modes**, meaning that the three remaining modes correspond to the raw uncompressed activation. In comparison, HOSVD allows us to retain an $\varepsilon$ fraction of the information **across all modes**. This explains why, in the same context, SVD generally offers higher accuracy but is more memory-intensive, while HOSVD is less accurate but more memory-efficient.

**#3 Compression overhead analysis (Answer for W2.1 - Q4 of HEKd and W1.1 of WgAz)**

In section A3 in supplementary, we will add description for overhead as follows:

"
**A.3 Details of Overhead, Complexity and Speed-up Computations**

Following the notation in our paper, we can compute the overhead, space complexity and the speed-up of HOSVD.

**Overhead:** During the forward pass, while Vanilla BP does not perform decomposition, HOSVD does. Therefore, the overhead of the training process is the computational complexity of HOSVD, which can be calculated as follows:

The computational complexity of SVD for a matrix of size $m \times n$ with $m \geq n$ is $\mathcal{O}_{\text{time}}(m^2n)$. HOSVD essentially performs SVD on each mode of the tensor. During the forward pass, at each convolutional layer, given the activation map of size $BCHW$, HOSVD involves performing SVD on four matrices of sizes $B\times CHW$, $C\times BHW$, $H \times BCW$, and $W\times BCH$. Therefore, the computational complexity for decomposition in the forward pass is:

$$ \begin{equation*}\mathcal{O}_{\text{time}}\left( \max(B, CHW)^2 \times \min(B, CHW) + \max(C, BHW)^2 \times \min(C, BHW) + \max(H, BCW)^2 \times \min(H, BCW) + \max(W, BCH)^2 \times \min(W, BCH)  \right) \end{equation*}$$

**Space complexity:** (refer to $complexity$ - formula (23) in our paper)

**Speedup:** (refer to $speedup$ - formula (24) in our paper)
"

Additionally, in Figure 2 in our paper we will add another sub-figure showing the predicted overhead as shown in Figure 3c of the PDF Rebuttal file.

**#4: Details on rank selection (answer for W3 - Q3 of dqZf and W1.2 of WgAz)**

We do not focus on rank but rather on how much information is retained after projecting tensors into a lower rank space, and we do that by manipulating the explained variance threshold $\varepsilon$, which is 0.8 or 0.9 in our experiments. We **are the first method** allowing for explicit control over the information loss. In lines 185 - 187, we wrote "The larger the explained variance, the closer $\tilde{A}$ will be to $A$, intuitively allowing for better estimation when performing backpropagation." We use the explained variance threshold $\varepsilon$ to control how much information can be retained after compression. This value goes from 0 (loss of all information) to 1 (retention of all information). Moreover, as shown in Figure 2c in our paper and Figure 2d in the PDF Rebuttal file , as $\varepsilon$ is closer to 1, $SNR$ becomes bigger resulting in improved performance.

---

### Comment · Area_Chair_WUjK · 2024-08-08
**Start reviewer-author discussions right now**

Dear reviewers,

Authors submitted rebuttals, which should be visible to you now.

Please read the rebuttals carefully and start discussions with the authors now.

The reviewer-author discussion period will end on August 13, 2024. Since authors usually need time to prepare for their responses, your quickest response would be very appreciated.

In case you had requested additional experiments / analysis and the authors provided in the rebuttal, please pay extra attention to the results.

Thank you,
Your AC

---

> ### Comment · Area_Chair_WUjK · 2024-08-13
> **Reviewer-author discussions will end in about 30 hours**
>
> Dear reviewers,
>
> This is the final reminder for the reviewer-author discussions.
> It will end on August 13 11:59 AoE, and then we will start AC-reviewer discussions.
>
> If you have already concluded the discussions with the authors, thank you so much!
>
> If you have not responded to the author rebuttal yet, please do so immediately. We have been waiting for your response.
>
> In case you missed it, the general author rebuttal includes a PDF file.
>
> Best,
> Your AC

---

### Decision · Program_Chairs · 2024-09-25

**Decision:**

Accept (poster)

**Comment:**

The authors submitted rebuttals, and all the reviewers here acknowledged them and had discussions with the authors.
I requested clarifications about their comments, and some of them responded.
Variance of the ratings was somewhat high as one of the reviewers (Reviewer HEKd) raised their score to 7, following multiple discussions with the authors. As they suggested, this is an interesting tensor decomposition research with great clarity and writing in general. They also value originality and its potential impact and interest of data center users.

I should clarify that I decided to put my focus on Reviewers HEKd, WgAz, and dqZf as I found none of the weaknesses claimed by Reviewer f1cQ's concrete, but more like questions. The reviewer argued that they had found authors' rebuttal satisfactory, but didn't provide any justifications (including their initial review) to support the rating.

Reviewer WgAz gave the lowest rating among the four reviewers, and key weaknesses they claimed in the initial review and follow-up are
1.  "why not search for a more appropriate model architecture using NAS?"
2.  "Rebuttal was not convincing because the experiments are on small datasets with small models. [1] is more efficient and doesn't affect model performance ".

For 1, I saw that the reviewer pushed NAS through interactions, but NAS is a resource-intensive approach. This work discusses on-device learning, and I believe that not using NAS cannot be a weakness.

For 2, I agree with the reviewer to the point in general, but again this work is about on-device learning. I also checked [1], but it doesn't discuss model performance (accuracy) but FLOPs, latency, and memory consumption.

Reviewer dqZf pointed out multiple weaknesses of this work, and the authors rebuttal addressed their concerns. But the reviewer didn't explain rationale of the rating 4. I requested a follow-up from the reviewer, but the reviewer did not answer my question.
Having reviewed the authors rebuttal, I found the rebuttal satisfactory. E.g. (1), "Severe performance degradation relative to memory reduction" should be observed as part of trade-off discussions, which also shows better trade-off by the proposed method than those of the baselines. E.g. (2), "Limited comparison with other activation map compression techniques" <- The authors provided additional experimental results using Swin Transformer model (a significantly different architecture) in the supplementary material, which seems like a reasonable choice to me. Overall, it seems to me that the reviewer is looking for reasons to reject.

Considering all these points, I found my position literally on on fence, but towards acceptance. My recommendation is based on the fact that the review by Reviewer HEKd shared very concrete suggestions and relevant concerns, and the authors successfully addressed the concerns through the discussions, thus I would like to recommend acceptance for this work.


[Suggestions]
While my recommendation is acceptance, I still highly recommend that the authors take into account all review comments and address their concerns in the camera-ready (e.g., clarifying motivation of this work).
- Replace "AI model" with "deep learning model". AI is not the right term in this scientific paper, but just a distraction.
- My recommendation is partly conditioned on the authors' code release.